# The Chance of Freezing – A conceptional study to parameterize temperature-dependent freezing by including randomness of INP concentrations

Hannah C. Frostenberg[1], André Welti[2], Mikael Luhr[3], Julien Savre[4], Erik S. Thomson[5], and Luisa Ickes[1]

[1]Department of Space, Earth and Environment, Chalmers University, Gothenburg 41296, Sweden
[2]Finnish Meteorological Institute, Helsinki 00101, Finland
[3]former Department of Meteorology, Stockholm University, Stockholm 10691, Sweden
[4]Meteorological Institute, Faculty of Physics, Ludwig-Maximilians-Universität, Munich 80333, Germany
[5]Department of Chemistry and Molecular Biology, University of Gothenburg, Gothenburg 41296, Sweden

**Correspondence:** Hannah Frostenberg (hannah.frostenberg@chalmers.se), Luisa Ickes (luisa.ickes@chalmers.se)

**Abstract.** Ice nucleating particle concentrations (INPCs) can spread over several orders of magnitude at any given temperature. However, this variability is rarely accounted for in heterogeneous ice nucleation parameterizations. In this paper, we present an approach to incorporate the random variation of INPC into the parameterization of immersion freezing and analyze this novel concept with various sensitivity tests. In the new scheme, the INPC is drawn from a relative frequency distribution of cumulative INPCs. At each temperature, this distribution describing the INPCs is expressed as a log-normal frequency distribution. The new parameterization scheme does not require aerosol information from the driving model to represent the heterogeneity of INPCs. The scheme's performance is tested in a large-eddy simulation of a relatively warm Arctic mixed-phase stratocumulus. We find that it leads to reasonable ice masses in the cloud, especially compared to immersion freezing schemes that yield one fixed INPC per temperature and lead to almost no ice production in the simulated cloud. The scheme is sensitive to the median of the frequency distribution and highly sensitive to the standard deviation of the distribution, as well as to the frequency of drawing a new INPC and the resolution of the model. Generally, a higher probability of drawing large INPCs leads to substantially more ice in the simulated cloud. We expose inherent challenges to introducing such a parameterization and explore possible solutions and potential developments.

## 1 Introduction

Clouds play an important role in Earth's energy balance by reflecting incoming sunlight and interacting with infrared radiation. The cloud phase influences the amount of a cloud's radiative effect, but more importantly, it determines whether the cloud has a warming or cooling effect. According to Matus and L'Ecuyer (2017), liquid clouds have a global net radiative effect at the top-of-atmosphere (TOA) of -11.8 W m$^{-2}$, whereas ice clouds exert a warming of 3.5 W m$^{-2}$ and mixed-phase clouds cause a net cooling effect of -3.4 W m$^{-2}$.

The ice crystals in mixed-phase clouds usually originate from heterogeneous ice nucleation where ice nucleating particles (INPs) are necessary to trigger the phase transition. Ice nucleates heterogeneously in the atmosphere at temperatures between

0 and approximately -38 °C. At temperatures below -38 °C, homogeneous ice nucleation occurs spontaneously and freezing can happen without INPs. Heterogeneous ice nucleation can occur in different so-called modes: immersion freezing, contact freezing, deposition nucleation, and condensation freezing (see Vali et al., 2015, for definitions). We focus on immersion freezing where an INP is immersed in a supercooled cloud droplet and initiates freezing at a specific temperature. Different aerosol types can act as INPs: mineral dusts, biological and combustion particles, etc. The probability that an aerosol initiates ice nucleation increases with decreasing temperature. Measurements of ambient INP concentrations (INPC [$m^{-3}$]) show that at a given temperature, INPC can vary by several orders of magnitude, over time and space. Examples of spatial INPC variability in different marine locations, from approximately $10^{-1}$ to $10^3$ $m^{-3}$ at -15 °C can be found in Welti et al. (2020). Temporal variability, for example, the annual cycle of INPC in the Arctic, was reported to be up to three orders of magnitude (Wex et al., 2019). But also within smaller time periods down to single days, the INPC can fluctuate by up to three orders of magnitude (Bigg, 1961). It has been found that the INPC at a specific temperature is log-normally distributed. Log-normal frequency distributions of INPC occurrence have been measured at several locations within different environments (e.g., Isaac and Douglas, 1971; Bertrand et al., 1973; Radke et al., 1976; Flyger and Heidam, 1978; Conen et al., 2017; Welti et al., 2018; Hartmann et al., 2019; Schrod et al., 2020; Li et al., 2022). Ott (1990) showed that when an aerosol concentration is observed to have a log-normally distributed occurrence this likely results because the aerosol was subject to a series of random dilutions subsequent to emission.

Several parameterizations that simulate cloud droplet freezing exist and they are based on different physical variables. Burrows et al. (2022) distinguish between aerosol-aware and -unaware parameterizations. One example of the latter is the formulation by Fletcher (1962) (F62); a scheme that is based on the observation that the average INP concentration increases exponentially with decreasing temperature for -10 > $T$ > -30 °C. F62 only requires temperature information to calculate INPC. The scheme by Niemand et al. (2012) (N12) is an example of an aerosol-aware parameterization; it describes immersion freezing based on the active site density of desert dust aerosols observed in laboratory measurements. N12 is valid for -12 > $T$ > -36 °C and requires temperature, the number of dust aerosols, and the average INP surface as input to determine INPC. Another example of an aerosol-aware scheme is the parameterization by Phillips et al. (2008) (P08) which represents immersion, contact, and deposition freezing on dust/metal aerosol, organic carbon, and biological INPs. P08 uses temperature, water vapor saturation with respect to ice, and aerosol concentrations of the four aerosol species to predict INPC. P08 is valid for 0 > $T$ > -70 °C (or lower). The parameterization of Khvorostyanov and Curry (2000) (K00) is based on classical nucleation theory (CNT), where each substance is assigned a characteristic contact angle between a nucleating ice cluster and the particle surface. The smaller the contact angle, the more ice-active a substance. K00 utilizes temperature, water vapor saturation with respect to ice, particle radius, and one contact angle per substance to calculate ice nucleation rates. One method of including particle-to-particle heterogeneity of INPs in parameterizations is to apply a distribution on the contact angle parameter in CNT to calculate the ice-activity of the INPs (e.g., Marcolli et al., 2007 or Wang et al., 2014). These schemes require temperature and aerosol radius as input to the parameterization and use different values for the mean and standard deviation of the contact angle distribution for different aerosol species.

There are three drawbacks that ice nucleation parameterizations can have: i. They can be computationally complex, i.e., require

detailed input from models regarding the aerosol type, size, and/or number concentration. ii. They can be limited to a specific temperature range. iii. Most of them fail to reproduce INPC variability, i.e., for one set of environmental conditions (temperature, humidity, aerosol type, and concentration) they yield a single fixed INPC value.

The lack of INPC variability in simulations compared to atmospheric observations emerges from non-represented INP types and sources, the large size of model grid boxes, and the use of bulk aerosol concentrations (e.g., dust, soot, biological) as input variables in parameterizations. Even if a model includes information on, e.g., the detailed size distribution of dust aerosol, it will not represent all the dust INP types (different minerals) and their variability in the atmosphere. To circumvent these drawbacks we developed a parameterization of immersion freezing (F23) that simulates observed INPC and their variability

while only using temperature as input variable. F23 is valid for the entire temperature range of immersion freezing ($0 > T > -38\,°C$). The INPCs returned by F23 are drawn randomly from a log-normal distribution of INPCs for each temperature, thereby capturing the natural INP variability, without requiring information about the present ice-active aerosol. The random drawing allows to represent the INP population by a distribution of INPCs, instead of, e.g., one single value as in F62. The log-normal INPC distributions F23 is loosely based upon have been observed in the maritime boundary layer (see Sect. 2). The main goal

of our study was to test the concept of including INPC variability in a parameterization scheme. We compare the behavior of random drawing from an INPC distribution instead of using fixed values only depending on, e.g., temperature.

We test the new F23 parameterization in the large-eddy simulation (LES) model MIMICA (MISU/MIT Cloud-Aerosol model, Savre et al., 2014) to investigate how it performs at producing ice in a simulated cloud in comparison to conventional parameterizations (diagnostic ice crystal number concentration, and F62). We analyze the sensitivity of the scheme on its characteristics

and implementation details. The simulations are initialized based on in-situ observations during the ASCOS (Arctic Summer Cloud Ocean Study) Arctic campaign and represent a mixed-phase stratocumulus cloud with in-cloud temperatures between approximately -7 and -10 °C (see Sect. 2.2).

## 2 Method

### 2.1 Formulation of the F23 parameterization

The basic idea for the new F23 parameterization is that INPCs have large variability, even on short time scales (Bigg, 1961), which we represent by drawing a new INPC from the given distribution at a certain frequency if immersion freezing occurs (i.e., $T \leq 0\,°C$ and cloud droplets are present). Drawing a new INPC mimics the evolution of the ice-active aerosol population, for example by replenishment or time-dependent freezing at each grid point. The latter means that drawing at each time step does not necessarily reflect a change in aerosol population from one time step to another, but a change in the present INP

population due to the activation of more INPs with time. The main goal of this study is to test the concept of randomly drawing from a distribution of INPCs instead of applying a deterministic value (e.g., the median of observed INPCs) as is done in other freezing parameterization schemes.

We use the conceptional distribution of INPCs shown in Fig. 1, derived from the extensive data sets of a long-term time series of INPC temperature spectra measured in the maritime boundary layer at Cabo Verde (Welti et al., 2018) and widely dispersed

ship-based observations (Welti et al., 2020). On average, the INPC field observations show a striking consistency of the temperature spectra's shape and variability over time and location. The same consistencies exist when compared with other long-term (e.g. Schrod et al., 2020) or composite INPC data sets (e.g. Petters and Wright, 2015).

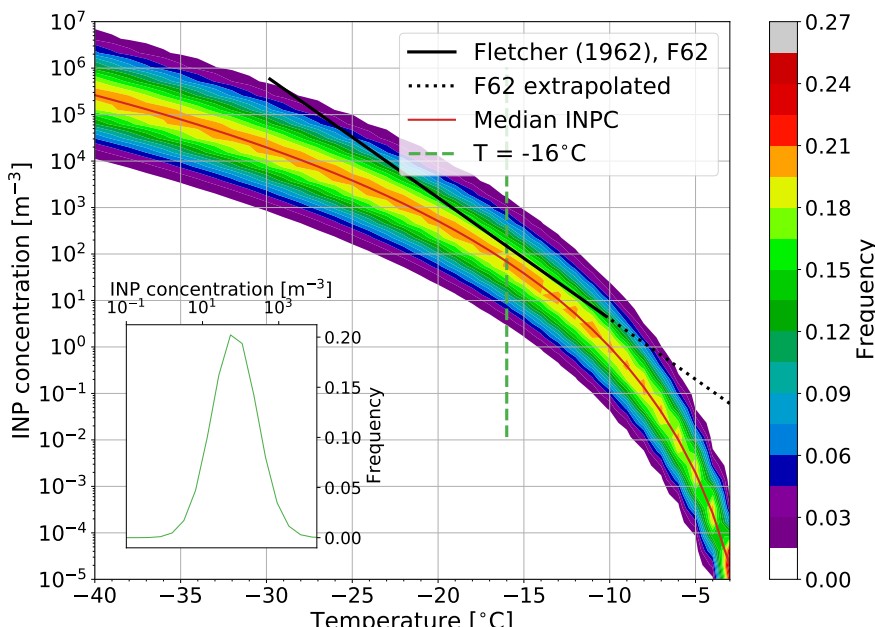

**Figure 1.** Relative frequency distribution (RFD) spectra for INPC as a function of temperature. Median INPCs are marked by the red line. The log-linear parameterization by Fletcher (1962) (F62) is shown within its validity range (black solid) and extrapolated to higher temperatures (black dotted) for comparison. The inset in the lower left corner shows the RFD at $T = -16\,°C$ (green dashed line).

Based on the aforementioned data sets, we derived a function for temperature-dependent, log-normally distributed INPC frequency:

$$D(\mu, \sigma^2) = \frac{1}{\sqrt{2\pi} \cdot \sigma} \exp\left(-\frac{[\ln(a \cdot \text{INPC}) - \mu(T)]^2}{2\sigma^2}\right), \tag{1}$$

with $a = 1\,\text{m}^3$, INPC in $\text{m}^{-3}$, $\mu(T)$ the temperature-dependent mean, and $\sigma^2$ the variance of the log-normal distribution (not the INPC itself). For the marine data sets we find $\sigma = 1.37$ and $\mu(T) = \ln(-(b \cdot T)^9 \cdot 10^{-9})$ with $b = 1/(1°C)$ and $T$ given in °C. By normalizing the distribution, a relative frequency distribution (RFD) as a function of temperature is obtained (Fig. 1). This normalization is necessary due to the discrete INPC field. Whenever an immersion freezing event occurs in the model

(i.e., $T \leq 0°C$ and cloud droplets are present), an INPC value is drawn randomly from the RFD. That is, for two grid points with freezing events at the same temperature, the drawn INPC can differ by several orders of magnitude. For a large number of grid points, the relative frequency of the drawn INPC will follow a log-normal function with the median INPC (red line in

Fig. 1) having the highest probability. For example, if all grid points were at a temperature of -16 °C, the frequency of the drawn INPC would follow the distribution shown as a green curve in the lower left Fig. 1 inset.

F23 assumes that all INPs are immersed in cloud droplets. To take into account the dynamic evolution of ice formation, the ice crystal number concentration ($N_i$) at a grid point is subtracted from the drawn INPC. This returns the number of newly frozen cloud droplets in a time step, i.e., hydrometeors moving from class *cloud droplet* to class *ice crystal*. Subtracting $N_i$ is one approach to solve the need for time discretization when implementing a deterministic time-independent scheme. Note that if other frozen hydrometeor species (snow, graupel) are represented in the model (not implemented in our setup), the sum of their

number concentrations should be subtracted from the drawn INPC. No negative tendencies (INPC-$N_i < 0$) are allowed in the scheme since already frozen cloud droplets will not melt due to a decrease in the INPC. The change in the respective mixing ratios is calculated by multiplying the change in the number of frozen droplets by the average cloud droplet mass:

$$\Delta N_i = -\Delta N_c = \max([\text{INPC} - N_i], 0) \,, \tag{2a}$$

$$\Delta Q_i = -\Delta Q_c = \Delta N_i \frac{\overline{Q_c}}{\overline{N_c}} \,. \tag{2b}$$

Here, $N_i$ and $N_c$ are the ice crystal and cloud droplet number concentrations in m$^{-3}$, $Q_i$ and $Q_c$ are the ice crystal and cloud droplet mixing ratios in kg m$^{-3}$, respectively, with the mean cloud droplet mixing ratio and number concentration denoted by the bar. The calculations are repeated at each time step, which means that INPs are only partially depleted through the subtraction of $N_i$. We interpret this as a way to imitate time-dependent freezing and INP recycling, which has been shown to be crucial for realistic cloud development in LES for Arctic mixed-phase clouds (e.g., Solomon et al., 2015).

The implemented INPC RFD-field is discretized into bins of INPC and temperature. The INPCs differ approximately by a factor of 2 (or $\Delta \log_{10}(\text{INPC}) \approx \log_{10}(2)$), while the temperature bins have a size of 1°C. For this reason, we define temperature to the nearest degree when drawing from the INPC RFD if not stated otherwise. Using the F23 immersion freezing scheme in MIMICA has approximately the same computational expense for the total simulation time as any other interactive ice nucleation parameterization, for example, F62.

### 2.1.1  Example of parameterized INP concentrations


To illustrate the INPC distribution from the F23 parameterization, let us assume that we have a uniform -16 °C cloud spreading horizontally over the entire model domain consisting of 1000 grid points. The INPC RFD at all cloud grid points is represented by a log-normal distribution curve (inset in Fig. 1). This results in an INPC of 68.7 m$^{-3}$ being drawn with the highest probability since this is the median INPC at -16 °C: Med[INPC($T = -16$°C)]$= \exp(\mu)$ m$^{-3} = -T^9 \cdot 10^{-9}$ m$^{-3} \approx 68.7$ m$^{-3}$. In the

model, 20.2 % of the grid points will draw the median INPC, that is 202 of the 1000 cloud grid points will get the median INPC of 54.3 m$^{-3}$ (differing from the theoretical value of 68.7 m$^{-3}$ because of discrete INPC binning). The range of INPC bins with a relative frequency > 0.1 % covers INPCs of 0.8 - 7543.1 m$^{-3}$, which means that rarely INPCs $\leq 0.8$ or $\geq 7543.1$ m$^{-3}$ will be drawn for the example cloud. If no ice was present previously, the number of cloud droplets frozen at this time step equals the INPC (Eq. 2a), limited by the total number of cloud droplets present.

### 2.1.2 Representing the RFD

Since the parameterization draws values from a distribution, it needs to be ensured that there are enough random draws in order to represent the distribution well: INPCs in the model should vary according to the distribution, but different model runs should also be reproducible. To investigate how many draws are necessary to represent the INPC distribution, we conducted several drawing tests (drawing, e.g., 50 times from the distribution vs. drawing 100 times) at -16 °C and compared the relative frequencies of the drawn values to the theoretical values by calculating the root mean square error (RMSE) between the drawn and theoretical distributions. Comparing the results of the different drawing tests suggests that 300 random draws lead to a reproducible prediction of the RFD (see Table 1 and Fig. 2); the RMSE converges for $\geq 300$ with a constant first derivative (linear slope) of the connecting lines, and the standard deviation decreases only slightly for $> 300$ draws.

The domain used in the simulations has $96 \times 96$ grid points in the horizontal, leading to 9 216 grid points in each layer. The simulated cloud is stratocumulus (see Sect. 2.2), where the temperature field is horizontally uniform. Assuming that at least one layer of grid points in the model has the same temperature due to the uniform structure of the stratus cloud, this translates into a minimum of 9 216 grid points with the same temperature. This implies a minimum of 9 216 draws from the RFD at one temperature, which is substantially more than the minimum number of draws determined to well-represent the distribution (300). Hence, we confirm that a representative INPC distribution is being drawn from the RFD. This has also been verified in MIMICA simulations (not shown).

**Table 1.** Average and standard deviation of RMSE for different draw amounts from the INPC RFD at -16 °C compared to theoretical RFD values. Each drawing test was performed 100 times and the table contains the average and standard deviation of the RMSE over these 100 times.

| Number of draws | Mean RMSE [$m^{-3}$] | Standard deviation RMSE [$m^{-3}$] |
|---|---|---|
| 50 | $17.8 \cdot 10^{-3}$ | $5.47 \cdot 10^{-3}$ |
| 100 | $12.4 \cdot 10^{-3}$ | $3.51 \cdot 10^{-3}$ |
| 200 | $8.65 \cdot 10^{-3}$ | $2.58 \cdot 10^{-3}$ |
| 300 | $7.28 \cdot 10^{-3}$ | $2.03 \cdot 10^{-3}$ |
| 400 | $6.26 \cdot 10^{-3}$ | $1.59 \cdot 10^{-3}$ |
| 500 | $5.67 \cdot 10^{-3}$ | $1.40 \cdot 10^{-3}$ |
| 600 | $5.27 \cdot 10^{-3}$ | $1.49 \cdot 10^{-3}$ |
| 700 | $4.77 \cdot 10^{-3}$ | $1.39 \cdot 10^{-3}$ |
| 800 | $4.45 \cdot 10^{-3}$ | $1.19 \cdot 10^{-3}$ |
| 900 | $4.18 \cdot 10^{-3}$ | $1.08 \cdot 10^{-3}$ |
| 1000 | $3.73 \cdot 10^{-3}$ | $1.11 \cdot 10^{-3}$ |

### 2.2 Simulation setup

We use the well-established large-eddy simulation (LES) model MIMICA (MISU/MIT Cloud-Aerosol model, Savre et al., 2014). For more information on the model, see additionally Appendix A or, e.g., Savre and Ekman (2015a), Savre and Ekman (2015b), or Sotiropoulou et al. (2020). The simulated case is based on a mixed-phase Arctic stratocumulus. The case-study

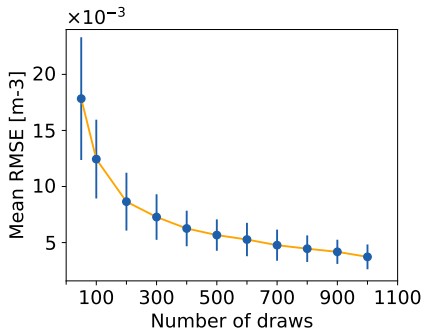

**Figure 2.** Average and standard deviation of RMSE for the drawing tests in Table 1 are decreasing for increasing number of draws at -16 °C.

stratocumulus cloud was observed between 30 August and 31 August 2008 during the ship-based ASCOS campaign (Arctic Summer Cloud Ocean Study; Tjernström et al., 2014). At that time, the research vessel Oden was drifting with an ice floe located at approximately 87° N. The atmospheric conditions were characterized by a high-pressure system with large-scale subsidence in the free troposphere (for details see Tjernström et al., 2012). This case has been used to study other microphysical cloud properties like dissipation (Loewe et al., 2017), the influence of CCN hygroscopicity on cloud properties (Christiansen et al., 2020), secondary ice production (Sotiropoulou et al., 2021) and sustenance (Bulatovic et al., 2021) of an Arctic mixed-phase cloud, as well as in a model intercomparison study (Stevens et al., 2018). We selected this case because it is an established case and because there are large uncertainties about the nature and concentration of INPs in the Arctic, which poses a challenge to modeling mixed-phase clouds in this region. Using other immersion freezing parameterizations in MIMICA, for example, an active site scheme following Ickes et al. (2017), to simulate this case requires unrealistically active INPs in order to form ice. We focus on immersion freezing in this study because liquid-dependent ice nucleation is dominant in Arctic stratiform clouds (de Boer et al., 2011). Contact freezing can be neglected since very little interstitial aerosol was present in the case. Furthermore, these aerosols would need to collide with the supercooled cloud droplets in a very stable non-turbulent cloud, making contact freezing unlikely to occur. Deposition nucleation has the same limitation when it comes to interstitial aerosol and can additionally be neglected because of the temperature range which is not favorable for deposition ice nucleation. Secondary ice formation is not explicitly taken into account in this study since we focus on primary ice formation. Simulations of the case with a diagnostic ice crystal number concentration (STD) instead of an interactive representation of ice nucleation assume a minimum ice crystal number concentration of 200 m$^{-3}$ at grid points with a temperature below 0 °C where there are sufficient cloud droplets. This means that at any given time step in the cloud if $N_i$ falls below 200 m$^{-3}$ and $T < 0$°C, ice crystals are produced to retain $N_i = 200$ m$^{-3}$, no matter the exact temperature below 0 °C. This approach is unspecific to the ice formation mechanisms. It combines immersion and contact freezing (due to the requirement of cloud droplets), as well as secondary ice processes.

The MIMICA simulations are initialized with the profiles of thermodynamic variables (e.g., potential temperature and pressure) and liquid cloud water measured at approximately 06 UTC on 31 August 2008. The initial ice/liquid potential temperature

profiles are randomly perturbed in order for convection to develop more quickly. Consequently, any two simulations will yield different results, even if all parameters are held constant. A cloud layer was present between ca. 550 and 900 m above ground level, capped by a temperature and humidity inversion and de-coupled from the surface (see Sotiropoulou et al., 2021 for profile details). The temperature within the cloud ranged from approximately -7 to -10 °C. The simulation setup follows Sotiropoulou et al. (2021). The domain covers a $96 \times 96 \times 128$ grid with a constant horizontal spacing of $dx = dy = 62.5$ m (6 km×6 km horizontal domain size). The vertical spacing is 7.5 m near the ground and in the cloud layer; between the surface and the cloud it changes sinusoidally and reaches a maximum $dz$ of 25 m, with a 1.7 km total vertical domain size. The time step is dynamic in order to satisfy the Courant-Friedrichs-Lewy (CFL) condition for the leapfrog time-integration method and ranges from $\approx$ 1 - 3 s to prevent numerical instabilities within the model. Our simulations cover 12 hours, with the first two hours utilized as a spin-up period and subsequently omitted from the results. A large-scale steady state is maintained throughout the model runs and the cloudy layer is present in the initial state of the simulations. However, the initial cloud is liquid and cloud ice is only formed from the first model time step.

We excluded the hydrometeor categories snow and graupel from all simulations since it is known that MIMICA produces rather large amounts of graupel for this case (Stevens et al., 2018) which dominates both ice water path (IWP) and the number concentration of frozen hydrometeors. Because we are primarily interested in ice formation in our study, ice crystals being the only frozen hydrometeors simplifies the analysis. For the same reason, snow and graupel were excluded previously from Arctic stratocumulus simulations conducted with MIMICA (Savre and Ekman, 2015b). Excluding snow and graupel means that no cold collection processes are active in our simulations, however, ice crystals can still grow by deposition, be transported, and precipitate. Aerosol is not represented prognostically in the model setup.

Observed liquid water path (LWP) and IWP were derived from measurements by a micrometer radiometer and millimeter cloud radar respectively (Tjernström et al., 2012). Uncertainties for LWP are 25 g m$^{-2}$ while for IWP they are approximately a factor of two (Shupe et al., 2013).

## 3    Results and discussion

We first compare the baseline version of the new F23 parameterization (F23, all parameters are set to the values described in Sect. 2.1) to the standard setup (STD, diagnostic ice crystal number concentration), available observations, and F62.

Domain-averaged liquid water path (LWP [g m$^{-2}$]), ice water path (IWP [g m$^{-2}$]), ice crystal number burden (ICNB [m$^{-2}$]) and net infrared radiation at the surface (NIRS [W m$^{-2}$]) for STD, F23, and F62 are shown together with the median and the upper and lower quartiles of observational values for LWP and IWP in Fig. 3. The model was run ten times using STD or F23, respectively. The different "ensemble members" are initialized by randomly perturbed profiles and are represented by their median, maximum, and minimum values in Fig. 3.

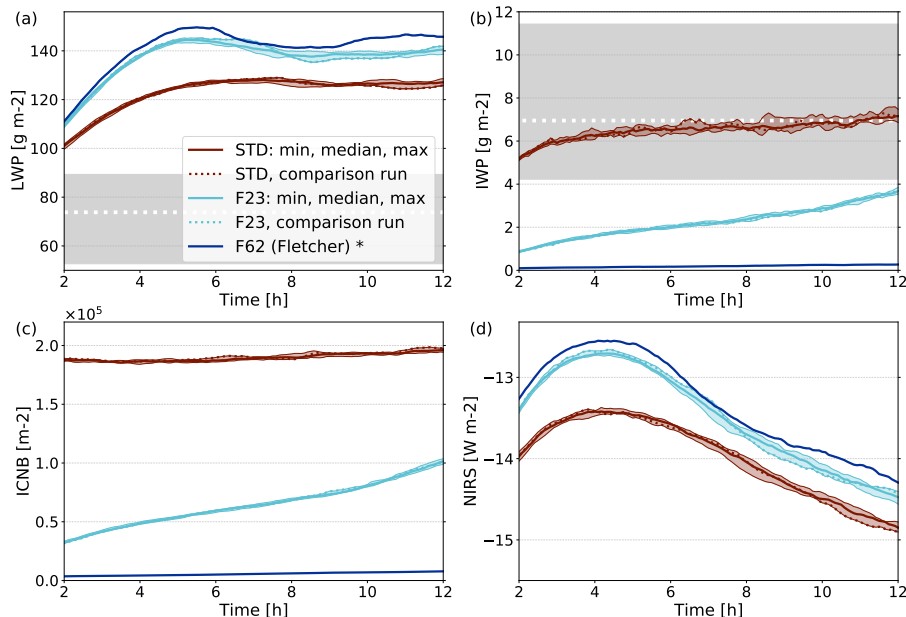

**Figure 3.** Domain-averaged **(a)** LWP, **(b)** IWP, **(c)** ICNB and **(d)** NIRS for 10 simulations of STD (red; median with min/max envelope) and with the F23 parameterization (cyan; median with min/max envelope). In both cases one representative simulation is plotted as a dotted line to be included in Figs. 4, D1, 6, 8, 10, 12, C1. The interquartile range of observations for LWP and IWP is indicated by the gray-shaded areas. Median observations are marked by the white dotted lines. One simulation with F62 is shown in blue. Significance was assessed with a two-sided Kolmogorov-Smirnov test at the 95 % level. Significant differences in IWP of F62 to F23 are indicated with an asterisk.

## 3.1 Comparison of F23 to STD (diagnostic ice crystal number concentration)

The F23 and STD simulations differ only in how ice crystal formation is parameterized. Other related processes like deposition and sublimation are treated in the same manner and the general setup follows Sect. 2.2. For F23, all parameters are set to the values described in Sect. 2.1, while STD ensures a minimum $N_i$ of 200 m$^{-3}$ where the temperature is below 0 °C and there are sufficient cloud droplets.

LWP is substantially larger for F23 than for STD (Fig. 3a). IWP is larger for STD than for F23 (Fig. 3b), but does not
compensate for the differences in LWP (the total mass of ice and liquid is still larger for F23). The large differences in LWP between F23 (INPC distribution) and STD (constant $N_i$) are due to enhanced freezing in STD, which leads to increased evaporation of liquid droplets and deposition onto ice crystals via the Wegener-Bergeron-Findeisen process (WBF), see Fig. B1. To analyze the ice crystal number, Fig. 3c shows ICNB, the vertically integrated $N_i$ (Fig. C1a). For STD, ICNB increases slightly from $1.88 \cdot 10^5$ m$^{-2}$ to $1.96 \cdot 10^5$ m$^{-2}$ over 10 hours, while it increases dramatically throughout the F23 simulations
from $3.2 \cdot 10^4$ m$^{-2}$ to $1.0 \cdot 10^5$ m$^{-2}$. Several primary processes cause the increase of ice mass and number: i. The vertical extent of the liquid cloud increases throughout the simulation (see dashed lines in Fig. 4). ii. The temperature within the cloud decreases throughout the simulation (see white contour lines in Fig. 4a and c). iii. For F23, new ice crystals form whenever

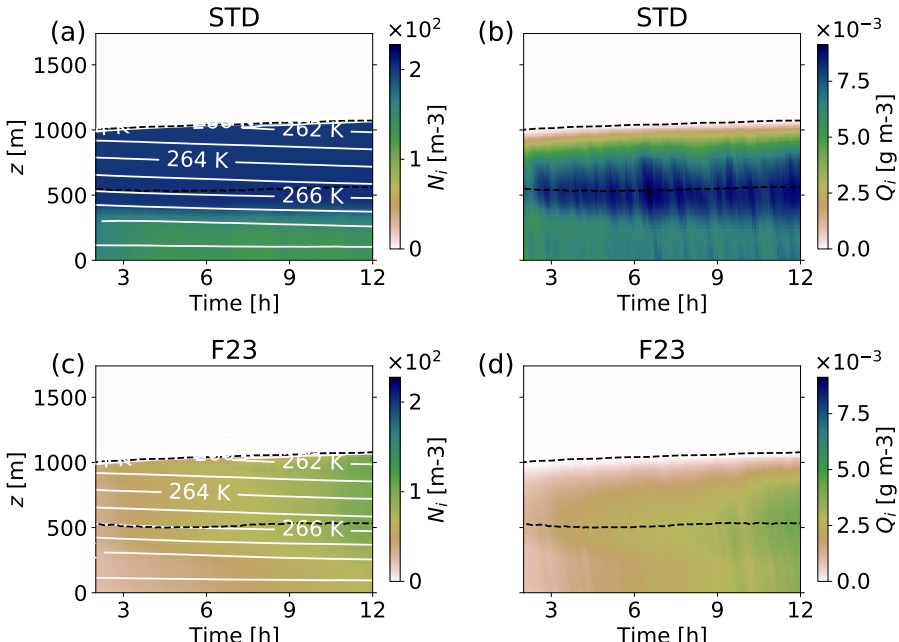

**Figure 4.** Domain-averaged profiles for simulation time after 2 hour spinup-time: **(a)** and **(c)** $N_i$ with contours of temperature, **(b)** and **(d)** ice mixing ratio ($Q_i$). The upper row (**a, b**) is the STD comparison run (see red dotted lines in Fig. 3), lower row (**c, d**) is the F23 comparison run (see cyan dotted lines in Fig. 3). The dashed lines show the cloud's top and bottom.

the drawn INPC is larger than the current $N_i$. Only process i. is relevant for STD since its requirements for ice formation are $T < 0°C$ and the abundance of cloud droplets. For F23, all three processes are relevant since it requires cloud droplets (i.), the

INPC RFD depends on the temperature (ii.), and iii. is an inherent characteristic.

Figure 3d shows the time series of NIRS, i.e., incoming minus outgoing infrared radiation at the surface. NIRS is larger for F23 compared to STD. This can be expected since LWP is larger for F23. The evolution over time is similar in both cases, with NIRS first increasing, then decreasing from hours 4-5. This pattern emerges from the combination of first increasing, then steady LWP (for F23, LWP decreases slightly between hours 5 and 8, which causes a larger decrease in NIRS), and

continuously decreasing temperature (see white lines in Fig. 4a and c) due to radiative cooling of the cloud.

The profile of $N_i$ for STD in Fig. 4a illustrates the principle of prescribing $N_i$ diagnostically: for STD, within the cloud and ca. 100 m below it, $N_i$ has the prescribed minimum value of 200 $m^{-3}$. Down to the surface, the concentration is lower but constant, which is caused by steady sedimentation of ice crystals from above combined with sublimation from ice crystals (see Fig. B1b). $Q_i$ has the maximum values at the cloud bottom (Fig. 4b), which is commonly observed in Arctic mixed-phase

clouds (Shupe et al., 2008). Since no cold-phase collection processes are active in these simulations, the mass of single ice crystals can only grow by deposition. The total mass of ice however is affected also by transport processes like sedimentation. The vertical distribution of $N_i$ and $Q_i$ in F23 is similar to STD (Fig. 4c and d). However, both $Q_i$ and $N_i$ increase by larger

rates throughout the simulation time for F23, which can also be seen in the IWP and ICNB values (Fig. 3b and c). Towards the end of the simulation time, F23 ice values are about half of STD. Even though ice nucleation depends on the temperature in F23, $N_i$ is quite homogeneously distributed throughout the cloud (Fig. 4c). We explain this by the uniform stratification of the cloud and the subtraction of $N_i$ from INPC (Eq. 2a) in the scheme, which homogenizes $N_i$ vertically over time.

## 3.2 Comparison to observations

LWP as calculated in MIMICA is substantially higher than the 75[th] percentile of the observed values, irrespective of the treatment of ice nucleation (Fig. 3a). Stevens et al. (2018) found this already in their model inter-comparison study, where MIMICA was amongst the models with the highest LWP. The LWP exceeded the observations for simulations similar to ours with no prognostic aerosol treatment but a simplified treatment of aerosol activation (note however, that the MIMICA-simulations in Stevens et al., 2018 included snow and graupel). MIMICA-simulations, where aerosol is modeled prognostically, yielded results closer to the observational range in the study by Stevens et al. (2018). The IWP for STD lies within the interquartile range of the observations (Fig. 3b) which was found in Stevens et al., 2018 as well. The minimum $N_i$ in STD was chosen in order to yield ice masses in the cloud similar to the observations. For the F23 simulations, IWP is lower than the 25[th] percentile of the observations. F23 however only represents immersion freezing, leading to a lower ice mass. We can expect that multiplication processes were present in the observed cloud, which would have led to a higher IWP compared to model simulations that do not represent SIP. In fact, Sotiropoulou et al. (2021) simulated the same case using MIMICA and including a description of ice multiplication from breakup during ice-ice collisions. Their results show an increase in IWP by a factor of 2 to 3 when breakup is activated in the model. Adjusting our simulated IWP accordingly would result in values corresponding to the observed IWP for F23. LWP is decreased by approximately 25-35 g m$^{-2}$ in simulations with ice multiplication in Sotiropoulou et al. (2021). This would bring our modeled LWP values closer to the observations, but LWP would still be higher than observed, due to simplified aerosol activation as explained above.

One aspect that complicates the comparison of our simulations with observations is that we excluded snow and graupel. If snow and graupel were included in the simulations, IWP might increase as riming converts liquid to frozen mass. On the other hand, this might also lead to faster precipitation and thus depletion of liquid or frozen water.

Note that our goal was to test F23 rather than include all necessary processes in order to closely model the observations.

## 3.3 Comparison of F23 to interactive F62 implementation

F62 calculates the number of cloud droplets that freeze to ice crystals from the temperature-dependent INPC [m$^{-3}$] given in Fletcher (1962) (and shown by the black line in Fig. 1):

$$\text{INPC}(T) = 0.02 \cdot \exp\left(-\beta \cdot T\right) \tag{3}$$

with $\beta = 0.6/(1°\text{C})$ and $T$ in °C. The changes in $N_i$, $N_c$ $Q_i$ and $Q_C$ are calculated according to Eq. 2. Note that this parameterization is only strictly valid for -10 $> T >$ -30 °C, but we extrapolate it to the temperatures of the simulated cloud (dotted line in Fig. 1). Analogously to F23, freezing happens where $T \leq 0°\text{C}$ and cloud droplets are present.

**Table 2.** Overview of sensitivity studies

| Simulation name | Abbrevia- tion | Frequency of drawing from RFD | Horizontal grid spacing | Minimum vertical grid spacing | Multiplica- tor of RFD Median | RFD Stan- dard devia- tion | Size of tempera- ture-bins |
|---|---|---|---|---|---|---|---|
| standard[a] | STD | - | 62.5 m | 7.5 m | - | - | - |
| baseline F23 | F23 | every $\Delta t$[b] | 62.5 m | 7.5 m | 1 | 1.37 | 1 °C |
| median 1.5 | M1.5 | every $\Delta t$ | 62.5 m | 7.5 m | 1.5 | 1.37 | 1 °C |
| median 1.25 | M1.25 | every $\Delta t$ | 62.5 m | 7.5 m | 1.25 | 1.37 | 1 °C |
| median 0.75 | M0.75 | every $\Delta t$ | 62.5 m | 7.5 m | 0.75 | 1.37 | 1 °C |
| median 0.5 | M0.5 | every $\Delta t$ | 62.5 m | 7.5 m | 0.5 | 1.37 | 1 °C |
| sigma 1.5 | S1.5 | every $\Delta t$ | 62.5 m | 7.5 m | 1 | 1.5·1.37 | 1 °C |
| sigma 1.25 | S1.25 | every $\Delta t$ | 62.5 m | 7.5 m | 1 | 1.25·1.37 | 1 °C |
| sigma 0.75 | S0.75 | every $\Delta t$ | 62.5 m | 7.5 m | 1 | 0.75·1.37 | 1 °C |
| sigma 0.5 | S0.5 | every $\Delta t$ | 62.5 m | 7.5 m | 1 | 0.5·1.37 | 1 °C |
| sigma 0 | S0 | - | 62.5 m | 7.5 m | 1 | -[c] | 1 °C |
| half degree | 0.5Deg | every $\Delta t$ | 62.5 m | 7.5 m | 1 | 1.37 | 0.5 °C |
| once 5 sec | 5S | once per 5 sec | 62.5 m | 7.5 m | 1 | 1.37 | 1 °C |
| once 10 sec | 10S | once per 10 sec | 62.5 m | 7.5 m | 1 | 1.37 | 1 °C |
| once 20 sec | 20S | once per 20 sec | 62.5 m | 7.5 m | 1 | 1.37 | 1 °C |
| once 5 min | 5M | once per 5 min | 62.5 m | 7.5 m | 1 | 1.37 | 1 °C |
| once 60 min | 60M | once per 60 min | 62.5 m | 7.5 m | 1 | 1.37 | 1 °C |
| delay[d] | D-10S-5M | -, once per 10 sec, once per 5 min | 62.5 m | 7.5 m | 1 | 1.37 | 1 °C |
| low resolution | F23 LR | every $\Delta t$ | 125 m | 15 m | 1 | 1.37 | 1 °C |

[a] Simulation with diagnostic $N_i$, no interactive ice nucleation parameterization.

[b] Time step

[c] In this case, no distribution is used, but the median of the distribution is used for all freezing cases.

[d] Simulation where ice nucleation is started after two hours, first with a drawing frequency of every 10 seconds, then after an additional two hours the drawing frequency is decreased to 5 minutes.

Figures 3b and D1 show that very little ice is produced using the F62 ice nucleation scheme. We explain this by the subtraction of $N_i$ from INPC($T$) since this only leads to considerable ice formation at the beginning of the simulation, when no ice is present yet, or when the temperature decreases. This comparison between F23 and F62 illustrates the difficulty of simulating reasonable ice masses in warm mixed-phase clouds with conventional schemes that yield one INPC for one set of environ- mental conditions. Drawing from a distribution of INPCs according to F23 leads to substantially larger ice masses, and these 275 might even be multiplied if, e.g., secondary ice processes would also be considered (see discussion in Sect. 3.2). In the future, it could be interesting to further analyze if the difference between F23 and F62 could be reduced by tuning F62 and how much it is related to the conceptual differences of the schemes.

### 3.4 Sensitivity studies of F23

To investigate the sensitivity of the simulated cloud to the characteristics of the F23 scheme, the following parameters (sum- 280 marized in Tab. 2) were varied: i. the median and standard deviation of the INPC distribution (Secs. 3.4.1 and 3.4.2); ii. the size of the temperature bins (Sect. 3.4.3); iii. the frequency of drawing (Sect. 3.4.4); iv. the resolution of the model domain

**Table 3.** The three quartiles (25$^{\text{th}}$ percentile: P$_{25}$, etc.) of IWP for all sensitivity studies for the final four simulated hours (8–12 h) and differences relative to F23 ($\Delta$F23) or STD ($\Delta$STD). Values in g m$^{-2}$. Simulations with significant differences to F23 (STD) are highlighted with an asterisk. Significance was assessed with a two-sided Kolmogorov-Smirnov test at the 95 % level. D-10S-5M is not included, because values are similar to 5M or 60M.

| Simulation | IWP P$_{25}$ | $\Delta$F23/ $\Delta$STD | IWP P$_{50}$ | $\Delta$F23/ $\Delta$STD | IWP P$_{75}$ | $\Delta$F23/ $\Delta$STD |
|---|---|---|---|---|---|---|
| F23 | 2.6 | - | 2.8 | - | 3.3 | - |
| M1.5 * | 3.7 | 1.1 | 4.0 | 1.2 | 4.8 | 1.5 |
| M1.25 * | 3.4 | 0.8 | 3.5 | 0.7 | 4.0 | 0.7 |
| M0.75 * | 2.0 | -0.6 | 2.2 | -0.6 | 2.4 | -0.9 |
| M0.5 * | 1.4 | -1.2 | 1.5 | -1.3 | 1.7 | -1.6 |
| S1.5 * | 16.8 | 14.2 | 17.3 | 14.5 | 18.1 | 14.8 |
| S1.25 * | 7.3 | 4.7 | 7.8 | 5.0 | 8.7 | 5.4 |
| S0.75 * | 0.9 | -1.7 | 1.0 | -1.8 | 1.2 | -2.1 |
| S0.5 * | 0.4 | -2.2 | 0.4 | -2.4 | 0.5 | -2.8 |
| S0 * | 0.1 | -2.5 | 0.1 | -2.7 | 0.1 | -3.2 |
| 0.5Deg | 2.6 | - | 3.0 | 0.2 | 3.3 | - |
| 5S * | 2.1 | -0.5 | 2.2 | -0.6 | 2.6 | -0.7 |
| 10S * | 1.8 | -0.8 | 1.9 | -0.9 | 2.1 | -1.2 |
| 20S * | 1.5 | -1.1 | 1.7 | -1.1 | 1.9 | -1.4 |
| 5M * | 1.2 | -1.4 | 1.3 | -1.5 | 1.5 | -1.8 |
| 60M * | 1.2 | -1.4 | 1.3 | -1.5 | 1.4 | -1.9 |
| F23 LR * | 1.8 | -0.8 | 1.9 | -0.9 | 2.3 | -1.0 |
| STD | 6.7 | - | 6.8 | - | 7.0 | - |
| STD LR * | 6.3 | -0.4 | 6.5 | -0.3 | 6.7 | -0.3 |

(Sect. 3.4.5). The sensitivity tests i. mean that we cover a wider spectrum of INPC RFDs than the one defined in Sect. 2.1 and shown in Fig. 1.

### 3.4.1 Median of the distribution

The median $(\exp(\mu))$ of the INPC RFD is multiplied by a factor (0.5-1.5) shifting the entire distribution in Fig. 1 up or down, without changing the standard deviation. Having more INPs (higher median) or fewer INPs (lower median) impacts the amount of ice formed in the cloud. The effect can be seen from the modeling results of IWP (Tab. 3; Fig. 5), when increasing/decreasing the median by 25 % or 50 %, IWP increases/decreases accordingly. The vertical profiles exhibit this symmetry even more clearly with linear $N_i$ and $Q_i$ increases/decreases (Fig. 6a and b) and all changes are significant. The

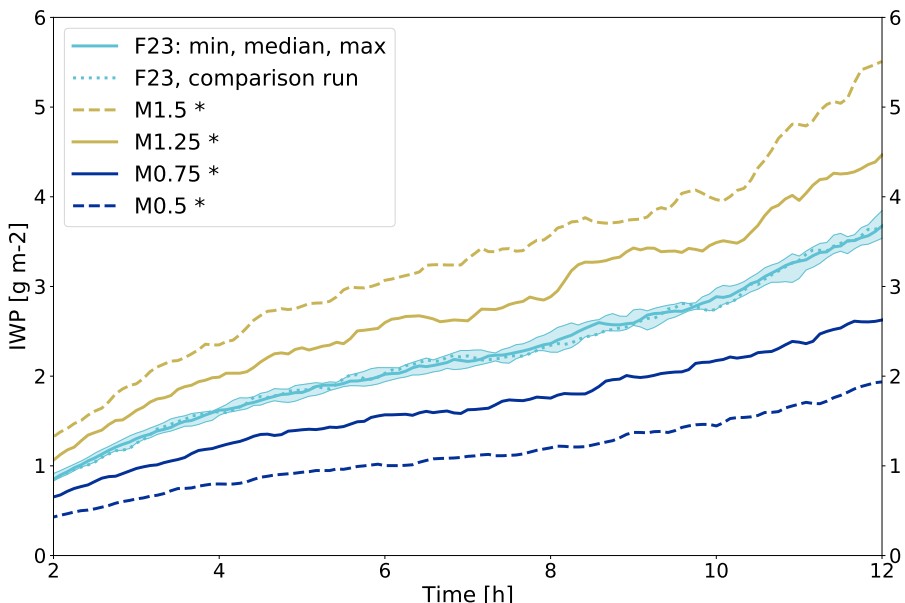

**Figure 5.** Domain-averaged IWP for 10 F23 simulations (cyan; median with min/max envelope, comparison run used in Fig. 6 is shown as a dotted line). Four sensitivity runs where the median of the RFD was changed (yellow: increased median INPC, blue: decreased median INPC). Significance was assessed with a two-sided Kolmogorov-Smirnov test at the 95 % level. Significant differences to F23 are indicated with an asterisk.

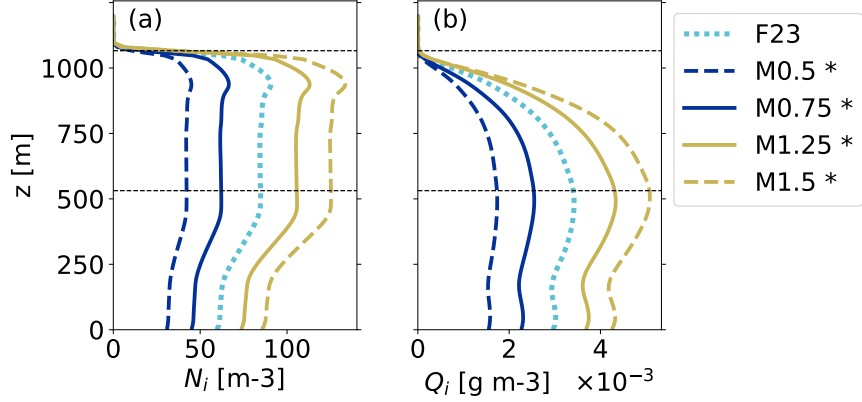

**Figure 6.** Profiles averaged over the domain and the simulation period of 8–12 hours: **(a)** $N_i$, **(b)** $Q_i$. The F23 comparison run is plotted in dotted cyan (see dotted line in Fig. 5), M0.5 and M0.75 in blue and M1.25 and M1.5 in yellow. Simulations with significant differences to F23 are indicated with an asterisk (tested with a two-sided t-test at the 95 % level). Horizontal dashed lines indicate the cloud's top and bottom in F23.

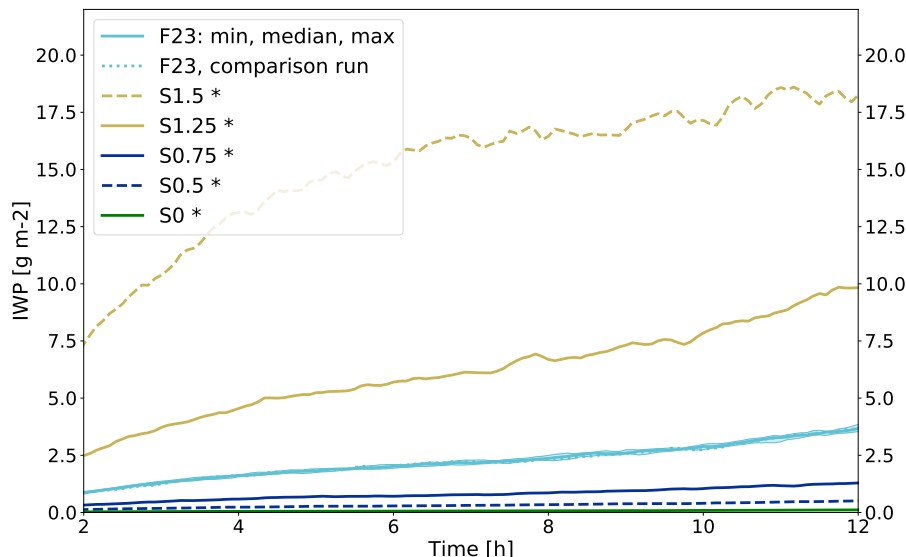

**Figure 7.** Domain-averaged IWP for 10 F23 simulations (cyan; median with min/max envelope, comparison run used in Fig. 8 is shown as a dotted line). Five sensitivity runs where the standard deviation of the RFD was changed (yellow: larger standard deviation, blue: smaller standard deviation, green: no standard deviation (only median, S0)). Significant differences to F23 are indicated with an asterisk.

significance of differences in IWP between simulations is tested with a two-sided Kolmogorov-Smirnov test at the 95 % level. Changes in vertical profiles are tested with a two-sided t-test at the 95 % level. It is expected that all variables concerning ice crystals increase (decrease) with increased (decreased) median INPC, since more (fewer) INPs lead to more (fewer) cloud droplets freezing. The change is linear because the median is logarithmized in the formula's exponent (see Eq. 1). No change in the vertical distribution of cloud droplet concentration is apparent (not shown).

### 3.4.2 Standard deviation of the distribution

The standard deviation ($\sigma$) of the RFD determines the variability of the INP concentration. The wider the distribution, the larger the variability of drawn INPCs. For the modeled cloud, a larger variability results in substantially and significantly increased IWP, $N_i$ and $Q_i$ while a lower variability results in significantly smaller values (Fig. 7 and Fig. 8). The changes are exponential with linear changes of $\sigma$ since $\sigma$ is in the exponent in the INPC RFD (Eq. 1). These results emphasize that it is the large INPCs that dominate ice formation in F23. Increased ice formation triggered by large INPCs could lead to cloud glaciation and subsequent cloud dissipation in colder clouds if ice crystals grow at the expense of liquid droplets due to the WBF process. In other simulations including secondary ice processes, for example, mechanical splintering or break-up of ice crystals (e.g., Field et al., 2016), a high $N_i$ can be relevant to even further enhance $N_i$. For example, Yano et al. (2016) report a critical $N_{i,\text{crit}}$ which can lead to an explosive enhancement of the ice crystal number. This $N_{i,\text{crit}}$ might only be reached if there is a possibility to draw large INPCs with F23. If large INPCs become less probable (smaller $\sigma$), IWP, $N_i$, and $Q_i$ decrease.

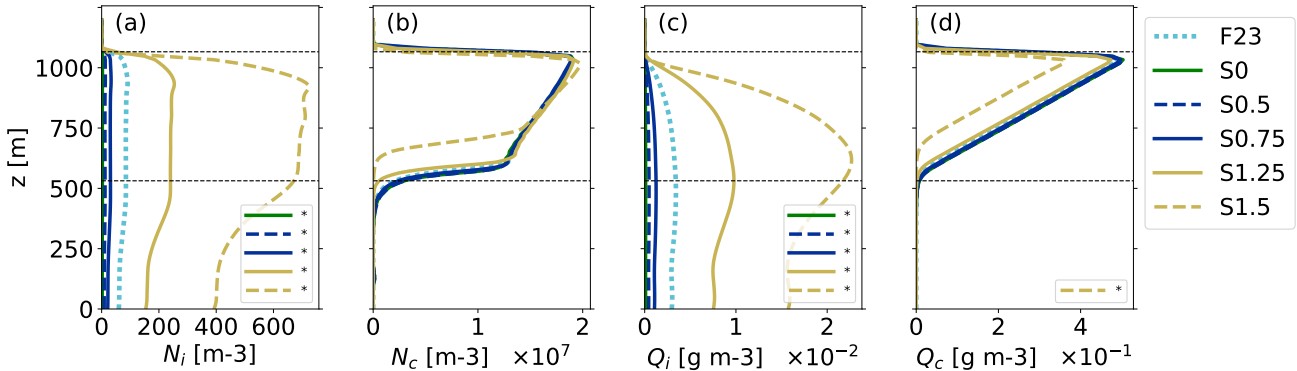

**Figure 8.** Profiles averaged over the domain and the simulation period of 8–12 hours: **(a)** $N_i$, **(b)** $N_c$, **(c)** $Q_i$ and **(d)** $Q_c$. The F23 comparison run is plotted in dotted cyan (see dotted line in Fig. 7), S0 in green, S0.5 and S0.75 in blue, S1.25 and S1.5 in yellow. Simulations that yielded significant differences to F23 are shown in the respective variable's legend in the subplots **(a)**-**(d)**. Horizontal dashed lines indicate the cloud's top and bottom in F23.

This is also apparent when analyzing the simulation without an INPC distribution, using the RFD's median value at all time steps (green line in Figs. 7, 8a and c). All ice variables (IWP, $N_i$, and $Q_i$) have the lowest values for this run (see Table 3). Using realistic median INPCs at the rather high temperature of the simulated case leads to almost no ice formation (see also Sect. 3.3 where the INPCs are much larger than the median of the F23 RFD, but almost no ice is produced). Once an INPC

distribution is added, considerable ice is formed. Excluding negative $\Delta N_i$ (Eq. 2a) causes $N_i$ to progressively increase even without decreasing cloud temperature. Such increases can be expected from time-dependent immersion freezing.

It is remarkable that the large increase in ice for S1.5 even leads to changes in $N_c$ and $Q_c$ (Fig. 8b and d). The cloud bottom is elevated by ca. 100 m in comparison to the F23 case (Fig. 8b). This is probably because more cloud droplets freeze, leading to a depletion of unfrozen cloud droplets. Within the cloud, $N_c$ is very similar for S1.5 and F23. However, $Q_c$ for S1.5 is

significantly smaller, indicating an enhanced WBF process resulting in liquid droplet evaporation.

### 3.4.3 Size of the temperature bins

For the F23 simulation, we draw from the INPC RFD defined on 1 °C temperature bins centered on whole degrees (i.e., -0.5 to -1.5 °C is the -1 °C bin). Thus a change in temperature from e.g., -10.4 to -10.6 °C shifts the INPC from the concentrations at -10 °C to the concentrations at -11 °C of the INPC distribution. Using smaller temperature bins, we expect the parameterization

to be less sensitive to small temperature changes and lead to smaller changes in the drawn INPC concentrations. This effect was tested by decreasing the temperature bin range from 1 °C to 0.5 °C (0.5Deg in Tab. 3). Figure 9 shows the IWP for a run with 0.5 °C temperature binning. Overall, IWP for the 0.5Deg simulation does not significantly differ from the F23 runs. Until hour 8, IWP is slightly below the F23 case and between hours 9 and 11, IWP is slightly larger. The averaged profiles for the

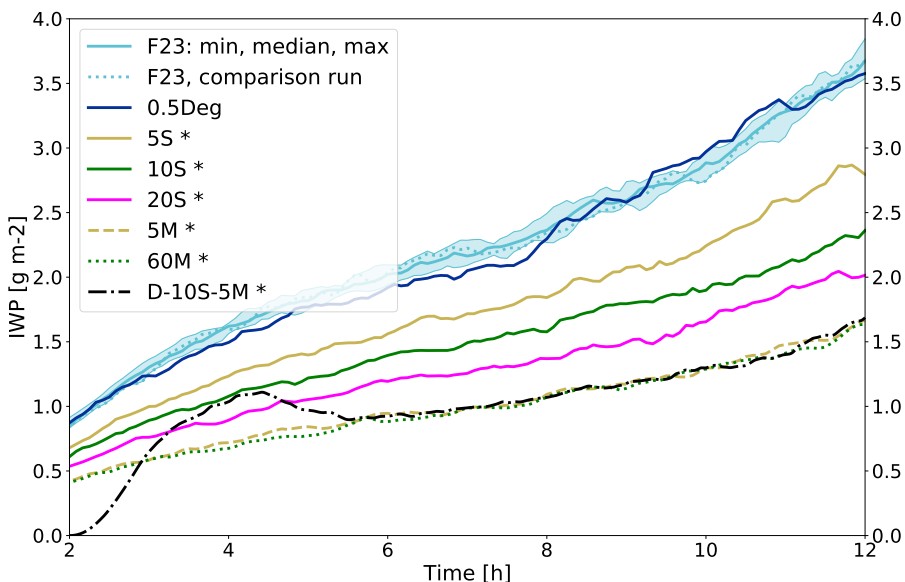

**Figure 9.** Domain-averaged IWP for 10 F23 simulations (cyan; median with min/max envelope, comparison run used in Fig. 10 is shown as a dotted line). Seven sensitivity tests (blue: 0.5Deg, yellow solid: 5S, green solid: 10S, magenta solid: 20S, yellow dashed: 5M, green dotted: 60M, black dash-dotted: D-10S-5M). Significant differences to F23 are indicated with an asterisk.

final four simulated hours are shown in Fig. 10. None of the variables differ significantly between F23 and 0.5Deg (see Tab. 3).
Judging from these results, there is no compelling benefit in decreasing the size of the temperature bins (from 1 °C to 0.5 °C).

### 3.4.4 Frequency of drawing

The frequency of drawing a new value for the INPC sets the length of the time period the drawn INPC is representative of the INPC at the grid point. The frequency of drawing is coupled to the model's temporal resolution since the maximum frequency of drawing is restricted by the time step of the model. The sensitivity to drawing frequency was tested by drawing the INPC
once every five seconds (5S, see Tab. 3), once every ten seconds (10S), once every 20 seconds (20S), once every five minutes (5M) and once every 60 minutes (60M) instead of at every time step (F23: every 1-3 seconds). Freezing events still occur at every time step (according to Eq. 2), but within the respective time period (e. g., five seconds or 60 minutes), the INPC is constant at one grid point.

IWP, $N_i$, and $Q_i$ for the runs with lower drawing frequencies exhibit similar relative increases over time as F23 but have
significantly lower absolute values (Fig. 9). The profiles of ice variables also decrease as the frequency of drawing decreases (Fig. 10). This evolution can be explained by the subtraction of $N_i$ at each time step. $N_i$ at a grid point will not change between time steps for which INPC $< N_i$. Only when a newly drawn INPC $> N_i$, new ice is formed at the grid point. An exception is if there is a sink for $N_i$ at the grid point (e.g., sedimentation, sublimation, or advection) and $N_i$ becomes smaller than INPC before the next draw of INPC since then INPC - $N_i$ cloud droplets will freeze. As the overall chance of drawing INPC $> N_i$

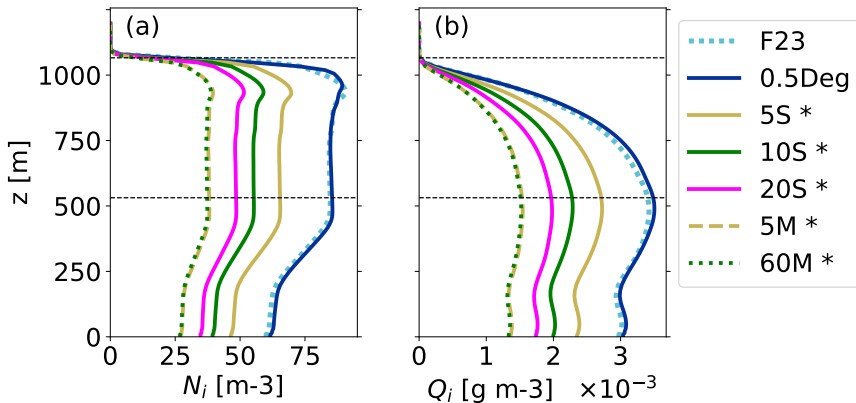

**Figure 10.** Profiles averaged over the domain and the simulation period of 8–12 hours: **(a)** $N_i$, **(b)** $Q_i$. One F23 run in dotted cyan (see dotted line in Fig. 9), 0.5Deg in blue, 5S in yellow solid, 10S in green solid, 20S in magenta solid, 5M in yellow dashed, 60M in green dotted. Simulations with significant differences to F23 are indicated with an asterisk. Horizontal dashed lines indicate the cloud's top and bottom in F23.

is higher with higher drawing frequency, the frequency of drawing new INPCs changes the amount of new ice formation and ice content in the cloud. Note that since large INPC and the chance to draw this large INPC determine the ice in the cloud, this introduces an indirect time-dependency of the scheme. The resulting ice in the cloud depends indirectly on the time until a large INPC value is drawn at a specific grid point. Increasing the drawing frequency leads to increases in the ice variables that do not appear to converge (Figs. 9 and 10, and Tab. 3). Ideally, one expects that the variables should converge for higher

drawing frequencies. This could possibly be achieved in future implementations of the scheme by introducing a correction factor that depends on the drawing frequency. As stated above, the maximum drawing frequency is the time step of the model, but the fact that the resulting cloud ice diverges with increased drawing frequency poses a limitation of F23. Note that this does imply that F23 depends on the time step of the model. In order to investigate the behavior of increasing drawing frequencies beyond what has been shown here, the time step of the model would need to be decreased further to be able to decrease the

drawing frequency. Even though this could theoretically be done, it would increase the computational costs tremendously and potentially other aspects of the simulation would change as well, which impedes a comparison. Therefore, we refrained from such a test. Another feature apparent in Figs. 9 and 10, as well as Tab. 3 is that the ice variables converge for drawing frequencies of five minutes or larger.

    The simulation D-10S-5M in Fig. 9 is a run where no ice nucleation is applied during the first two hours of the simulation. After

that, F23 is called with a drawing frequency of 10 seconds for two hours, and for the final eight hours, the drawing frequency was set to five minutes. The simulation is not included in Fig. 10 and Tab. 3 since it results in the same values as 5M or 60M. D-10S-5M shows that the current drawing frequency determines the amount of ice in the cloud.

    It is important to investigate how the ice content reacts on very high and low drawing frequencies (asymptotic behavior). In

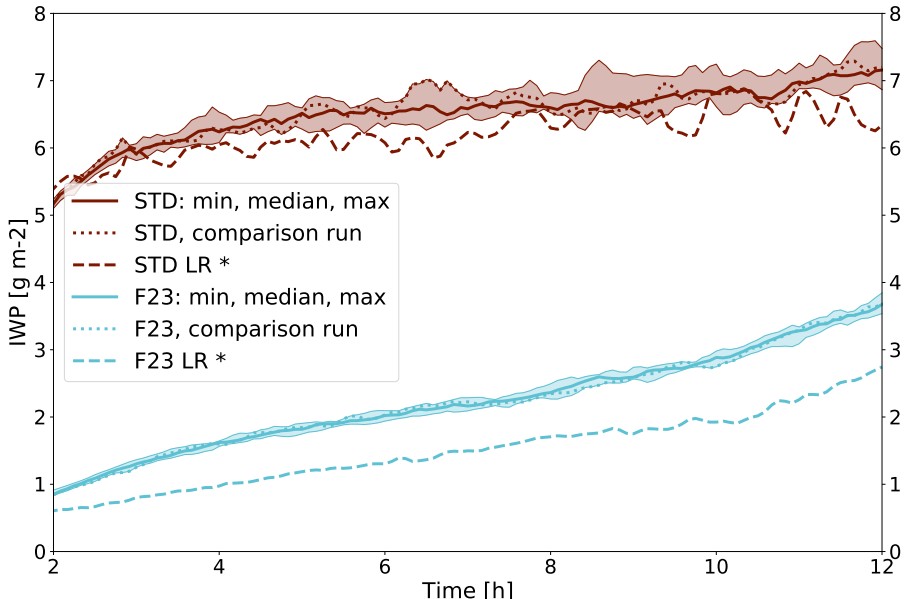

**Figure 11.** Domain-averaged IWP for 10 STD (red; median with min/max envelope, comparison run used in Fig. 12 is shown as a dotted line) and 10 F23 simulations (cyan; median with min/max envelope, comparison run used in Fig. 12 is shown as a dotted line). One run with half of the entire resolution respectively (dashed lines). Significant differences to STD or F23 are indicated with an asterisk.

this study, however, we are constrained by the model setup. Yet, we can expect that processes other than heterogeneous ice
nucleation would limit ice production for high drawing frequencies in more realistic model setups. For example, ice multi-
plication would lead to a quick increase in $N_i$, which would result in small or absent ice nucleation due to the subtraction of
$N_i$ according to Eq. 2a. We assume that the converging behavior of the ice mass for low drawing frequencies (no difference
between 5M and 60M) is caused by other components of the model than the heterogeneous ice nucleation scheme. If there
were no other processes affecting ice number, decreasing the drawing frequency can be expected to lead to decreasing ice in
the cloud.

### 3.4.5 Resolution of the domain

The spatial resolution of the model domain affects the F23 scheme in a similar way as the frequency of drawing, by changing
the number of draws of INPC for the same cloud. However, a change in the model resolution impacts all parts of the model,
including microphysical processes. The sensitivity of the LES at a lower resolution is tested by doubling the grid spacing in all
three dimensions while the domain volume remains constant (STD LR and F23 LR). Note that this will also lead to a doubling
of the model time step since it is calculated dynamically to satisfy the CFL criterion. Testing the scheme on a more coarse grid
is important because one goal for the new parameterization is that it can be adapted to larger-scale models where the resolution
will be much coarser than in large-eddy simulations. Lowering the resolution leads to significantly lower IWP for both the

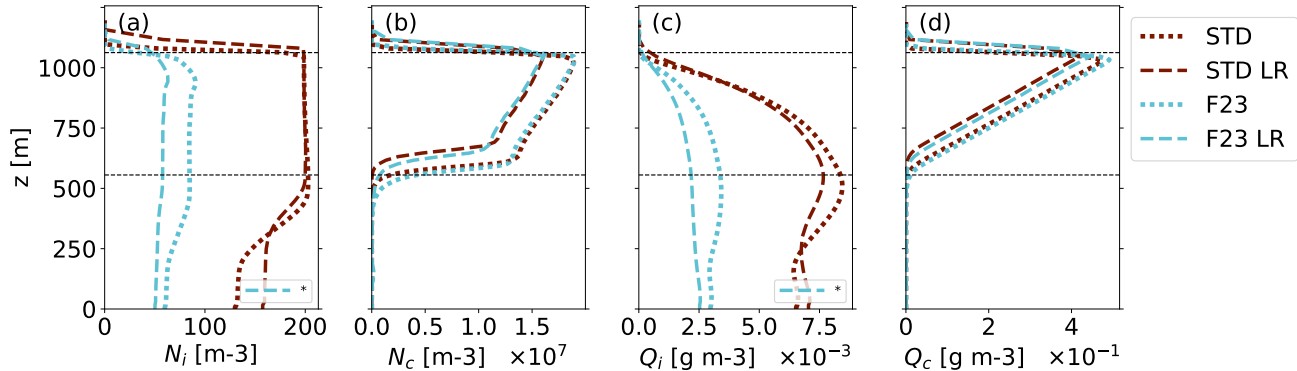

**Figure 12.** Profiles averaged over the domain and the simulation period of 8–12 hours: **(a)** $N_i$, **(b)** $N_c$, **(c)** $Q_i$ and **(d)** $Q_c$. One STD in red dotted, one F23 run in cyan dotted (see dotted lines in Fig. 11), STD LR in red dashed and F23 LR in cyan dashed. Simulations F23 LR in **(a)** and **(c)** yielded significant differences to F23. Horizontal dashed lines indicate the cloud's top and bottom in STD.

STD LR and F23 LR simulation (Fig. 11). IWP is affected more in the simulation with the F23 scheme. The vertical profiles
of the cloud droplet variables (Figs. 12b and d) show non-significant, but consistent decreases in $N_c$ (Fig. 12b), as well as $Q_c$
(Fig. 12d) for both runs with lower resolution. The decrease might be caused by changes in mixing processes, e.g., entrainment
at the cloud borders, due to the change in grid size. $N_i$ within the cloud is fixed to 200 m$^{-3}$ for STD LR and STD (Fig. 12a)
as per definition (see Sect. 2.2). Nevertheless, $Q_i$ is lower for STD LR compared to STD (Fig. 12c). This can be explained by
the decrease in $Q_c$ and thus a less efficient WBF process providing a smaller mass flux from liquid droplets to ice crystals.
For F23 LR, all variables shown in Fig. 12 decrease - for $N_i$ and $Q_i$ these changes are significant. The decrease in $N_i$ can be
explained by the smaller number of grid points in the domain as well as a coarser temporal resolution. INPCs will be drawn at
only one-eighth of the number of grid points and only about half as often in F23 LR and thus large INPCs will be drawn less
often than for F23. The results shown in Figs. 11 and 12 underline the conclusions from Sect. 3.4.2 and Sect. 3.4.4 that the
possibility to draw very large INPCs has the strongest effect on overall $N_i$.

## 3.5   Limitations and challenges

The limitations and challenges of this study can be divided into three aspects: limitations related to the parameterization itself
and the dataset it is based upon and limitations due to the modeling setup used to test the parameterization scheme.

**Limitations of the parameterization scheme**

Using a scheme with a random component, in our case the random drawing of INPC values, can introduce difficulties when
it comes to the technical implementation of such a scheme. Here, we assumed independence between consecutive random
draws from the INPC RFD, but a certain degree of autocorrelation both in time and space might need to be added to the

parameterization. Another challenge is the high sensitivity of simulated cloud ice to the INPC RFD's standard deviation, the drawing frequency, and the model resolution, i.e., the possibility to draw large INPCs. This aspect is partially related to the general difficulty in the creation of immersion freezing schemes where deterministic measurements have to be time-discretized. This leads to the challenge of selecting the best parameters for F23. Further investigations and constraints for the parameters in F23 are needed.

### Limitations related to the dataset the parameterization scheme is based upon

We base our parameterization's RFD on an extensive field-based dataset. However, this does not necessarily represent a globally valid INPC distribution. Since our study focuses on the conceptual idea of using an INPC distribution for parameterizing immersion freezing, we considered this limitation to be of secondary concern. Additionally, we present sensitivity tests with adjusted distributions that cover a large space of INP concentrations, i.e., the whole range of possible INPC. When using this scheme in the future, region or source-based RFDs can be applied. Another limitation is that the underlying INPC measurements in this study are surface-based and we assume the boundary layer to be well-mixed, and thus concentrations to be similar at cloud level. It is still an outstanding question (independent of this study) how surface-based measurements of INPs can represent INPC in the cloud-layer and be used for general freezing parameterizations. Finally, INP concentrations might change in a future climate and differ from the INPC RFD assumed here. However, it is easy to use an adjusted distribution in F23.

### Limitations of the modeling setup used to test the parameterization scheme

The model used for testing the F23 scheme, MIMICA-LES, and the chosen case study, ASCOS, are on their own a limitation of the study. The presented results and sensitivities are obtained by this model and based on this one case study, which results in further necessary investigations of F23.

More specifically, the choice of F23's INPC distribution is a limitation. While we simulate an Arctic cloud in this study, F23's INPC distribution is based on measurements at Cabo Verde and other maritime locations. However, Arctic INP measurements (e.g., from Ny-Ålesund in Wex et al., 2019) are covered by our INPC RFD. As mentioned above, another limitation is that F23's INPC distribution is based on surface-based measurements. However, in our case study, the cloud is actually de-coupled from the surface, thus surface INPC may not be representative of INPC in the Arctic cloud-layer (e.g., Creamean et al., 2018; Griesche et al., 2021).

Another limitation when it comes to the modeling setup, especially the comparison of the simulation results to observations, is the general setup of the model, i.e., the representation of microphysical processes, etc. However, we like to stress that the comparison to observations is not the purpose of this study.

## 4  Conclusions

A novel parameterization of immersion freezing that takes into account the observed variability of ice nucleating particles is presented. The observed INPC variability is reproduced by random drawing from an INPC relative frequency distribution that

depends on temperature. This means that the INP population at a specific grid point is represented by a new value at the chosen frequency of drawing. If the newly drawn INPC exceeds the present $N_i$, additional ice is formed. The F23 parameterization is valid for the entire temperature range of heterogeneous immersion freezing between 0 and -38 °C. F23 has the additional advantage that it does not require information on the present bulk aerosol from the atmospheric model, which makes it easy to implement and use in many different models. The main goal of this study was to test the general approach of representing immersion freezing by random drawing from an INPC distribution, as opposed to using traditional parameterizations that yield one INPC for the given temperature like, e.g., F62 (Fletcher, 1962).

We tested the parameterization for the large-eddy simulation of a mixed-phase Arctic stratocumulus cloud case with MIMICA. We used this case as a test bed for F23 because aerosol characteristics in the Arctic are largely unknown and this lack of information on aerosol bulk properties makes it challenging to use "classical" aerosol-aware freezing schemes. Moreover, it might be important to consider the whole range of possible INPCs in the Arctic summer where warm mixed-phase clouds are frequently observed. The simulations with F23 lead to less cloud ice than was observed, but our model setup does not include graupel and snow as well as ice nucleation modes other than immersion freezing, or secondary ice processes. The latter has been shown to increase IWP by a factor of 2-3 leading to the observed values in MIMICA-simulations of the same case by Sotiropoulou et al. (2021). Through our sensitivity tests, we found that the simulated IWP, $N_i$, and $Q_i$ of the cloud depend linearly on the median of the RFD that describes the INPC variability, and exponentially on the distribution's standard deviation. The large dependence on the standard deviation of the distribution is especially interesting, as it implies that the amount of ice in the modeled cloud is particularly sensitive to large INP concentrations. The possibility of drawing large INPC can influence cloud glaciation for colder cases caused by the WBF process and secondary ice processes when $N_{i,\mathrm{crit}}$ is reached (Yano et al., 2016). The relevance of randomly drawing INPC from a distribution is highlighted by the much lower IWP, $N_i$, and $Q_i$ when applying the Fletcher parameterization F62 or simply using the median INP concentration (i.e., no INPC distribution). This emphasizes that it is the rare, but large INPCs that control the freezing in the cloud, rather than the median INPC values. Additionally, the frequency of drawing a new INPC has a significant impact on the ice variables since it is when a new, larger INPC is drawn that additional ice is formed (Eq. 2a). The higher the frequency of drawing, the higher the ice content in the cloud. However, drawing every five minutes or less often (e.g., every 60 minutes) results in a similar amount of ice in the cloud. The high sensitivity of simulated cloud ice to an increased possibility to draw large INPC values (as tested by the increased standard deviation of the RFD, high drawing frequency, and higher spatial and temporal model resolution) poses the challenge of choosing the parameters (RFD standard deviation, and drawing frequency) for F23 while being in accordance with INPC observations. Further investigations are necessary to specify these parameters in order to apply the parameterization to other cases or in other models (including different model time steps). The unexpected divergence of the cloud ice amount for increased frequencies of drawing needs to be examined as well.

The scheme's independence from aerosol information in the atmospheric model is a strength but can be a limitation. The proposed INPC distribution of F23 may not be representative of distinct scenarios, e.g., a Sahara dust outbreak. The RFD would need to be updated with one that is based on INPC observations specific to an area or event of interest. However, the parameterization is flexible and can be adapted to different INPC observations. To represent several sources or source

regions at the same time in a model, different RFDs could be used depending on the location (remote vs. continental INP as an example). The independence to modeled aerosol might require updating the RFD when simulating future scenarios including

changes in aerosol or INP concentrations. Some degree of autocorrelation between subsequent random draws from the INPC RFD in time and space could be added to the scheme in the future. However, it is not clear what degree of autocorrelation would be physically reasonable. Since our study is based on one rather warm Arctic stratocumulus case, the parameterization should be tested and validated for other conditions and case studies in the future, especially for lower in-cloud temperatures. Additionally, F23 should be tested in LES-models other than MIMICA and in larger-scale models. The intention of this study

was to illuminate what including randomness in INPC leads to. We here show that representing INP heterogeneity in an immersion freezing parameterization allows for a realistic simulation of an Arctic stratocumulus cloud, but are clearly left with remaining challenges.

In order to make it easier to use INPC observations in schemes like the one presented, measurements should be reported either as individual measurement points (e.g., as time series) or, if aggregated, with the average and standard deviation and percentiles

of the underlying log-normal distribution.

*Code and data availability.* Implementation and sampling code as implemented for MIMICA can be obtained by contacting the corresponding authors. The model output data presented in this study is available at https://zenodo.org/ under the doi: 10.5281/zenodo.7572148.

*Author contributions.* **Idea**: AW, LI **Prestudy**: ML, AW, LI **Conceptualisation**: LI, HF **Model implementation**: HF **Simulations**: HF **Analysis of the data and visualisation**: HF, LI **Writing**: HF **Review & editing**: All authors.

*Competing interests.* The authors declare that no competing interests are present.

*Acknowledgements.* H.C.F. and L.I. were supported by Chalmers Gender Initiative for Excellence (Genie). E.S.T. was supported by the Swedish Research Councils, FORMAS (2017-00564) and VR (2020-03497), and the Swedish Strategic Research Initiative Modelling the Regional and Global Earth system, MERGE. The computations were enabled by resources provided by the Swedish National Infrastructure for Computing (SNIC) at National Supercomputer Centre (NSC) partially funded by the Swedish Research Council through grant agreement

no. 2018-05973. Hamish Struthers at LiU is acknowledged for assistance concerning technical and implementational aspects in making the code run on the Tetralith resources. We thank Annica M. L. Ekman for the valuable discussions and two anonymous reviewers for their constructive comments. We used the colormaps provided by Crameri et al. (2020). The research presented in this paper is a contribution to the strategic research area MERGE.

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

## Appendix A: MIMICA LES model description

The MIMICA LES solves a system of non-hydrostatic anelastic equations and represents cloud microphysics through a two-moment bulk microphysics scheme. The prognostic variables are number concentrations and mass mixing ratios of up to five represented hydrometeors: cloud droplets, raindrops, cloud ice, snow, and graupel. All hydrometeor mass distributions are regular gamma distributions and the hydrometeors' terminal fall speeds are calculated from simple power laws dependent on
their diameters. Warm microphysical processes are modeled according to Seifert and Beheng (2001) and Seifert and Beheng (2006), while collection processes involving frozen hydrometeors and resulting in the hydrometeors sticking together follow Wang and Chang (1993). MIMICA represents the following cold collection processes: i. riming of ice crystals and graupel by cloud droplets (resulting in graupel); ii. riming of ice crystals, graupel, and snowflakes by raindrops (resulting in graupel); iii. autoconversion of ice crystals to snow; iv. self-collection of snowflakes; v. collection of cloud droplets by snow (resulting in
snow); vi. growth of a snowflake by aggregating ice crystals; vii. collection of snow by graupel (resulting in graupel). Cloud condensation nuclei (CCN) activation is described following Khvorostyanov and Curry (2006), which is based on a simple power-law depending on modeled supersaturation and a prescribed background CCN concentration. It is possible to describe aerosol prognostically in MIMICA including activation as CCN. The supersaturation is solved pseudo-analytically following Morrison and Grabowski (2008). In the standard version of MIMICA, primary ice formation is represented by maintaining
a constant ice crystal number concentration ($N_i$) within the cloud. However, it is also possible to choose an interactive ice nucleation scheme. Radiation is calculated according to Fu and Liou (1993). The initial ice/liquid potential temperature profiles are randomly perturbed in order for convection to develop more quickly. Consequently, any two simulations will yield different results, even if all parameters are held constant.

## Appendix B: Averaged Q-tendency profiles for STD vs. F23

Figure B1 shows tendencies relevant to the Wegener-Bergeron-Findeisen process (WBF), averaged over the domain and entire simulation time for STD and F23: condensation (positive values) and evaporation (negative values) for liquid water, $Q_l = Q_c + Q_r$, in Fig. B1a and deposition (positive values) and sublimation (negative values) for ice crystals, $Q_i$, in Fig. B1b. More deposition and evaporation occur in STD, illustrating the WBF process with a mass transfer from liquid to frozen water.

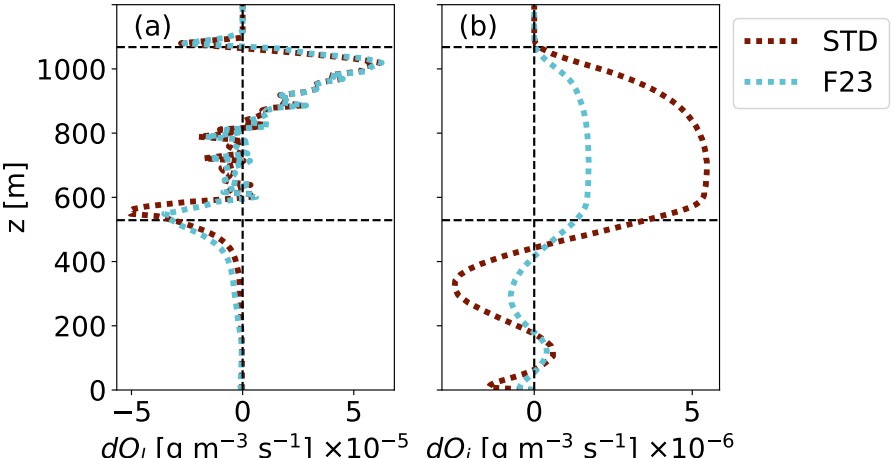

**Figure B1.** Profiles of WBF tendencies averaged over the domain and the entire simulation period: **(a)** evaporation/condensation, $dQ_l = dQ_c + dQ_r$, **(b)** sublimation/deposition, $dQ_i$. One STD in red dotted, and one F23 run in cyan dotted (see dotted lines in Fig. 3). Horizontal dashed lines indicate the cloud's top and bottom in F23 during the last four hours of the simulation.

## Appendix C: Averaged N-profiles for STD vs. F23

Figure C1 shows time- and domain-averaged profiles of the number concentrations of ice, cloud droplets, and raindrops for STD and F23.

## Appendix D: Averaged $N_i$ and $Q_i$ profiles for F23 vs. F62

Figure D1 shows time- and domain-averaged profiles of the number concentration and mixing ratio of ice for F62 compared to F23.

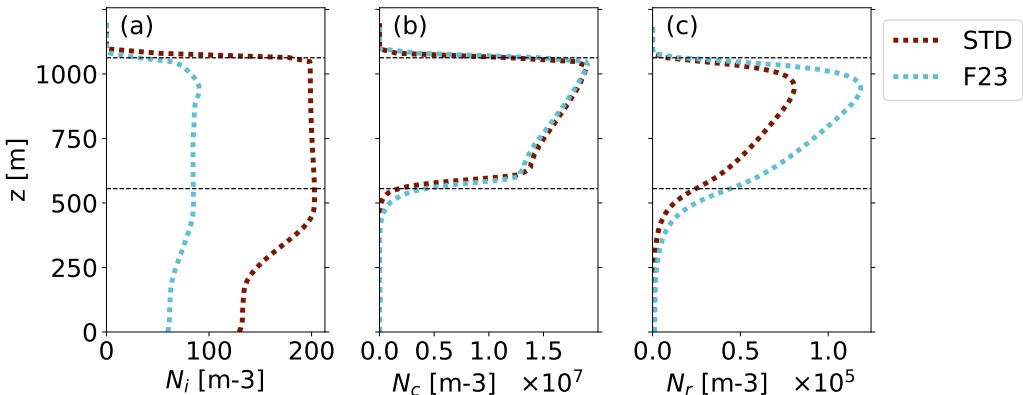

**Figure C1.** Profiles averaged over the domain and the simulation period of 8–12 hours: **(a)** $N_i$, **(b)** $N_c$ and **(c)** $N_r$. One STD in red dotted, and one F23 run in cyan dotted (see dotted lines in Fig. 3). Horizontal dashed lines indicate the cloud's top and bottom in STD.

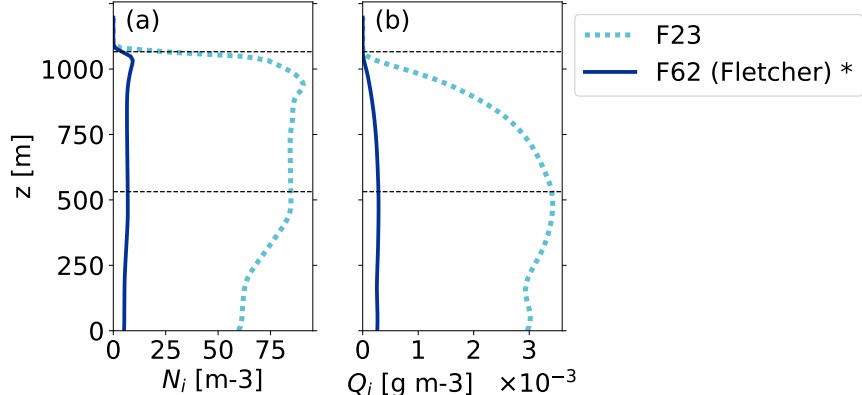

**Figure D1.** Profiles averaged over the domain and the simulation period of 8–12 hours: **(a)** $N_i$, **(b)** $Q_i$. The F23 comparison run is plotted in dotted cyan (see dotted line in Fig. 3), and one run with F62 in blue. Simulations with significant differences to F23 are indicated with an asterisk (tested with a two-sided t-test at the 95 % level). Horizontal dashed lines indicate the cloud's top and bottom in F23.