# Peer review of "The Chance of Freezing – A conceptional study to parameterize temperature-dependent freezing by including randomness of INP concentrations"

_Atmospheric Chemistry and Physics, 2022_

## Referee Comment (RC1)

**Review of "The Chance of Freezing – Parameterizing temperature dependent freezing including randomness of INP concentrations" by Frostenberg et al.**

**General comment:**

This study presents a new parameterization for heterogeneous cloud ice nucleation, producing INP concentrations as a function of temperature only. Previous parameterizations of INPs as a sole function of temperature (such as the one of Fletcher, 1962) produce a single value of INP concentration for a given temperature T, in order to reproduce the median INP=f(T) relationship observed globally and the overall temperature dependence of cloud ice nucleation. However, the approach of Fletcher and similar models fail to reproduce the very wide range of INP observed globally at a given temperature. The parameterization proposed in this study uses a stochastic approach, by including a range of INPs that can be sampled randomly from a frequency distribution around the central INP=f(T) value, so that a sufficiently large sample from the parameterization reproduce the observed range.

I found this idea interesting and worthwhile to explore. The authors have presented their results clearly and with a lot of care to explore sensitivities and impacts. Unfortunately, unless I am very mistaken, the results presented in the manuscript clearly show that this approach does not work when coupled to a cloud model and that it should not be used in such models, so I cannot recommend it for publication in ACP. Details are given below under "Major comments".

I realize that a large amount of work has been done for this study, and I actually found that understanding why this approach fails was very illuminating for understanding how INP parameterization works and their effects on clouds (see for example the comments on lines 231 and 260, and Major comment 1). I also believe such negative results are very useful for future research and should not be harder to publish than positive results (see for ex. Matosin et al., 2014). For this reason, I think that this study still be valuable to publish with a major overhaul of the manuscript and a change of title to clearly present these teachings of this experiment.

**Major comments:**

To clarify, I think that there are 4 major issues with the approach.

1) There is I think a fatal flaw in the stochastic approach of the parameterization, which is apparent in the high sensitivity of your results to the drawing frequency, and in the divergence of IWP at high drawing frequencies (Section 3.1.4 and Figure 9). If the drawing frequency is increased, the INP concentration should be more representative of the observed INP variability, which should lead to more accurate results if the parameterization works as intended.
On the opposite, results in Figure 9 show that increasing the frequency leads to arbitrarily high IWP, which does not even converge toward a given value at higher frequencies. On the opposite, the IWP actually converges for low drawing frequencies, when drawn INPs are less representative of the underlying distribution. Figure 9 only shows frequencies down to 0.2 Hz. However, with the study's approach, I am convinced that increasing the frequency further would produce arbitrarily high ice formation, and a high enough frequency would cause 100% freezing of all supercooled cloud droplets.
The reasons for this behavior are partly discussed in the text, and to clarify I will present my understanding here. Large INP concentrations in the atmosphere lead to cloud ice nucleation if the thermodynamic conditions for ice formation are present: for deposition freezing, if the air is

supersaturated with respects to ice; for immersion and contact freezing, if there are supercooled liquid droplets present. Once formed, ice crystals are thermodynamically stable (as opposed to supersaturated air or supercooled liquid) and they do not disappear if INP concentrations decrease. As briefly mentioned in the manuscript, his "INP memory" of ice crystals explains the sensitivity to both the standard deviation of the frequency distribution and the drawing frequency. Since the lognormal distribution is unbounded, any sufficiently large sample from the distribution will contain any arbitrarily large value of INP concentration INPC>Nc that can cause total supercooled droplet freezing. High drawing frequencies correspond to a larger number of samples that are more likely to contain high values, and large standard deviation means that drawing a large INP value is more likely for a given sample size.

Unfortunately, this is not only due to the unbounded lognormal distribution. Using a bounded frequency distribution would solve this issue either, because then a high enough drawing frequency would produce INPC values from the upper bound of the distribution (instead of indefinitely large values for the lognormal case). This would be equivalent to a Fletcher-like parameterization, but using this arbitrary new upper bound of the distribution instead of the median, with, in my opinion, no added value from the stochastic approach, but an additional complexity, increased computational cost, and lower reproducibility.

Conversely, do not I think using a very low drawing frequency makes sense. As mentioned above, it would mean that less representative INP concentrations are required for the parameterization to work, which defeats its whole purpose of better representing the observed INP range. In addition, using a very low frequency means that in a given grid cell the INP concentration is random (possibly very high or very low) but does not change at all over the drawing period, which is also very unrealistic. This would mean that a single grid point in the model domain will for example always freeze all droplets (or freeze none) during the whole frequency time, which is much larger than the advection time scale, during which different air masses or clouds can pass through this permanently high-INP (or low-INP) grid cell.

2) The parameterization represents the global and annual INP variability for a given T, modeled as a random process, and reproduces this variability at all times and locations. However, I think that this observed variability arises largely from non-random local (temporal and spatial) factors. For example, INP concentrations are almost always much lower in the coastal Arctic (Wex et al., 2019, cited in the manuscript) than near the equator (for example the Cabo Verde data used in this study, Welti et al., 2018, also cited already). In the coastal Arctic, INP concentrations are also much lower in winter than in summer, and are probably even lower in the middle of the ice pack where no identified INP sources exist. Even in Cabo Verde, where the data used in this study originates, Welti et al., 2018 mention that the high time variability of INPs is partly explained by emission sources and meteorological contexts. The issue here is that the approach presented in the present study can produce very high INP concentrations where and when they should be very low, for example during the Arctic winter. It is true that the Fletcher parameterization which also be biased in conditions that deviate significantly from median conditions. However, in the parameterization of Fletcher, the bias on INP concentrations will be limited by the fact that the median INP was used to build the parameterization. In the present study, extremely high INPs can be predicted at sites where they should be extremely low, something that does not occur when using relationships like Fletcher. In low INP cases, this is made worse by the issue discussed above in Major comment 1).

3) Unless I am mistaken, the parameterization of Fletcher predicts the total number of INPs, not just immersion mode INPs (it is based on observations of primary cloud ice, when INPC=Ni the

cloud ice crystal number). While the exact details of what is included in the STD control simulation are not clear, the present study seems to replace the Fletcher parameterization in the model, or a ice=f(T) relationship, by an immersion-mode-only parameterization. How are other INP processes (deposition, contact) parameterized in the model? This could also explain why the median INP=f(T) value on Figure 1. is much lower than in Fletcher, and, if other nucleation modes are completely ignored, why the new parameterization produces less ice than STD.

4) The abstract mentions that "reasonable ice masses" are produced by the model, but no comparison with observations of the new model is provided in the paper. Section 2.2 suggests that the original model underestimated cloud ice unless unrealistic INP concentrations were used, but the updated parameterization (F22) produces even less ice, so is it not degrading the model performance?

**Other comments**

1) The parameterization is said to be produced from observation data at Cabo Verde and on ships, but the comparison between observations and the parameterization is never shown. Can you add observed INP data on Figure 1 or add it as a supplementary figure?

2) L.49-50: I think this is slightly misleading because some of the observed variability can actually be explained by these meteorological and aerosol parameters. Complex models do not reproduce the full observed variability of course, but that is true of any model of any quantity. In addition, as mentioned in the introduction, the models of Marcolli et al., 2007 and Wang et al., 2014 do parameterize some of this enhanced variability.

3) L. 54: There is also an issue here, because the distribution presented in the paper does sample from the global variability of INPs, but is probably worse at representing the local (spatial and temporal) variability of INPs than aerosol-aware models, which for example will predict lower INPC over very remote regions where INPs are rare, and higher INPC over deserts where dust sources are plentiful.

4) L.73-74: Isn't this in contradiction with the discussion above of how variable INPs are in time and location? These observations used to build the parameterization should also be shown alongside the parameterization or in a supplementary figure.

5) Equation(1) – Why does the formula for μ gives negative values for T>-10C? Is there a typo here? I also think all the terms should be defined with units explained, is INPC supposed to be in #/m3?

6) L. 80 What do you mean by "normalizing the distribution"? What quantity is used to normalize it and why? A lognormal distribution should also in theory already be normalized. I suppose the issue could be that equation 1 is missing a 1/INPC factor in order to be truly lognormal, unless the distribution was meant to be defined in units of dD/dlog(INPC) (since dlog(INPC)/dINPC = 1/INPC).

7) L. 82 si>0 is only explicitly required for deposition freezing. For immersion freezing, there needs to be supercooled liquid droplets present (T<0., sw>0. and liquid droplets present). In practice, when the system is in equilibrium, si will always be positive where supercooled liquid exists, but this is not directly required for immersion freezing to occur.

8) L. 87-98. How are other nucleation modes treated? Even for immersion freezing only, INPs can also cause freezing of rain so why not also take into account the concentrations snow and other frozen precipitation when subtracting from INPC?

9) L. 96-98: A new drawing from the distribution will regenerate INPs, so I think that if the drawing frequency is smaller or equal to the time step this cycling does not occur. In addition, does this

mean that the drawing frequency, when larger than the time step, is also assumed to be the INP replenishment time in the cell by e.g. advection and resuspension?

10) L. 103-111 – Instead of discussing representativity with numbers of grid points for a given hypothetical cloud, I think it would be more clear to simply present this discussion as the representativity sample size of 1000 points.

11) 2.1.1 and 2.1.2 Since the sampling of the distribution is done independently in each grid point, INP concentrations can be extremely different in a grid point and the ones directly next to it, but it is probably not the case in reality, where the temporal scale of an aerosol plume can be several hours or even days, and the spatial scale several 10s of kms or even 100s of kms (for example Saharan dust events).

12) L.128 – Is the doubling of computing time 1) for just F22 compared to Fletcher, 2) for the microphysical calculation only or 3) for the total simulation time? If 3), how can you explain such a large cost from only adding a random sampling calculation to the model?

13) L. 130 Does the standard MIMICA (STD simulation) include Fletcher nucleation or is it with constant $N_i$, or something else? Description of the model in 2.2 indicates that $N_i$ can vary since it is corrected if falling above 200/m3.

14) L. 144 – Can secondary ice production in the cloud explain these ice biases, or has this been ruled out?

15) L. 162 – How was graupel and snow removed and can clouds precipitate or be removed by another process? Can this removal cause issues, for example, if there is no precipitation in the model have you checked that the cloud droplets and ice crystals do not grow to unrealistic sizes?

16) 2.2 and l. 171-172 – Can you explain more explicitly what are the differences between STD and F22, is this just the ice nucleation or are there other changes to how cloud ice is handled? Is graupel and snow also removed from STD?

17) 2.2 and Figure 3. - You mention that the issue in STD is the difficulty in reproducing observed cloud ice, but the IWP is even lower in F22, is this not degrading model results?

18) L. 181 - Can you provide a number estimate or figure showing this latent heat or temperature increase in the model?

19) L. 183- Can you also plot the water vapor column to show this gas-phase increase? The water budget should be balanced and it should be possible to show this shifting of water around phases directly.

20) L. 185 – Unfortunately, this increase with time is probably at least partly caused by the issues with the parameterization discussed above, because with enough time, INPC values large enough to freeze any supercooled droplet concentration will be sampled from the distribution. It seems that when discussing the sensitivity to drawing time the slope of the increase is indeed larger for smaller drawing times and the gap between experiments grows during the simulation.

21) L. 186 Can you also calculate here directly this effective cloud ice crystal mass from model results of ice crystal numbers and ice mass concentrations?

22) L. 231- 232 - This is very interesting and probably true for other parameterizations. For this reason, a parameterization only based on a median of INP observations will probably underestimate primary ice formation and some sort of correction might be necessary to consider short term variability and the likelihood of efficient INPs deviating from the median values. However, I am not sure of how applicable to other approaches this is. For example, earlier simple parameterizations like the one from Fletcher were I think directly based on observed ice crystal numbers, and did not suffer from this issue directly. And I think that some of the more complex aerosol-aware parameterizations will assume that these efficient INPs are quickly removed from the atmosphere and will not continue to nucleate ice later.

23) L. 260-261 – Is there a quantitative estimate of INPC variability at very short time scales (a few minutes and meters) in these references? INP are often observed at very low resolutions (very few observation sites, daily resolution or lower) so I am not sure how large this small-scale variability is, compared to the large-scale variability used to build the parameterization.

**References:**

Natalie Matosin, Elisabeth Frank, Martin Engel, Jeremy S. Lum, Kelly A. Newell; Negativity towards negative results: a discussion of the disconnect between scientific worth and scientific culture. Dis Model Mech 1 February 2014; 7 (2): 171–173. doi: https://doi.org/10.1242/dmm.015123

---

## Author Response (AR1)

**The Chance of Freezing – Parameterizing temperature dependent freezing including randomness of INP concentrations**

Hannah C. Frostenberg[1], André Welti[2], Mikael Luhr[3], Julien Savre[4], Erik S. Thomson[5], and Luisa Ickes[1]

[1]Department of Space, Earth and Environment, Chalmers University, Gothenburg 41296, Sweden
[2]Finnish Meteorological Institute, Helsinki 00101, Finland
[3]former Department of Meteorology, Stockholm University, Stockholm 10691, Sweden
[4]Meteorological Institute, Faculty of Physics, Ludwig-Maximilians-Universität, Munich 80333, Germany
[5]Department of Chemistry and Molecular Biology, University of Gothenburg, Gothenburg 41296, Sweden

We thank both reviewers for the constructive feedback. Below we copied the reviewers' comments in black and added our responses in blue.

Since a new calendar year has started, we now refer to the parameterization as F23.

**1   Response to review 1**

5   **General comment**

This study presents a new parameterization for heterogeneous cloud ice nucleation, producing INP concentrations as a function of temperature only. Previous parameterizations of INPs as a sole function of temperature (such as the one of Fletcher, 1962) produce a single value of INP concentration for a given temperature T, in order to reproduce the median INP=f(T) relationship observed globally and the overall temperature dependence of cloud ice nucleation. However, the approach of Fletcher and

10   similar models fail to reproduce the very wide range of INP observed globally at a given temperature. The parameterization proposed in this study uses a stochastic approach, by including a range of INPs that can be sampled randomly from a frequency distribution around the central INP=f(T) value, so that a sufficiently large sample from the parameterization reproduce the observed range.

I found this idea interesting and worthwhile to explore. The authors have presented their results clearly and with a lot of care

15   to explore sensitivities and impacts. Unfortunately, unless I am very mistaken, the results presented in the manuscript clearly show that this approach does not work when coupled to a cloud model and that it should not be used in such models, so I cannot recommend it for publication in ACP. Details are given below under "Major comments".

I realize that a large amount of work has been done for this study, and I actually found that understanding why this approach fails was very illuminating for understanding how INP parameterization works and their effects on clouds (see for example the

20   comments on lines 231 and 260, and Major comment 1). I also believe such negative results are very useful for future research and should not be harder to publish than positive results (see for ex. Matosin et al., 2014). For this reason, I think that this study

still be valuable to publish with a major overhaul of the manuscript and a change of title to clearly present these teachings of this experiment.

**1.1 Major comments**

25  1) There is I think a fatal flaw in the stochastic approach of the parameterization, which is apparent in the high sensitivity of your results to the drawing frequency, and in the divergence of IWP at high drawing frequencies (Section 3.1.4 and Figure 9). If the drawing frequency is increased, the INP concentration should be more representative of the observed INP variability, which should lead to more accurate results if the parameterization works as intended.

On the opposite, results in Figure 9 show that increasing the frequency leads to arbitrarily high IWP, which does not
30  even converge toward a given value at higher frequencies. On the opposite, the IWP actually converges for low drawing frequencies, when drawn INPs are less representative of the underlying distribution. Figure 9 only shows frequencies down to 0.2 Hz. However, with the study's approach, I am convinced that increasing the frequency further would produce arbitrarily high ice formation, and a high enough frequency would cause 100% freezing of all supercooled cloud droplets.

35  The reasons for this behavior are partly discussed in the text, and to clarify I will present my understanding here. Large INP concentrations in the atmosphere lead to cloud ice nucleation if the thermodynamic conditions for ice formation are present: for deposition freezing, if the air is supersaturated with respects to ice; for immersion and contact freezing, if there are supercooled liquid droplets present. Once formed, ice crystals are thermodynamically stable (as opposed to supersaturated air or supercooled liquid) and they do not disappear if INP concentrations decrease. As briefly men-
40  tioned in the manuscript, this "INP memory" of ice crystals explains the sensitivity to both the standard deviation of the frequency distribution and the drawing frequency. Since the lognormal distribution is unbounded, any sufficiently large sample from the distribution will contain any arbitrarily large value of INP concentration INPC>Nc that can cause total supercooled droplet freezing. High drawing frequencies correspond to a larger number of samples that are more likely to contain high values, and large standard deviation means that drawing a large INP value is more likely for a given sample
45  size.

Unfortunately, this is not only due to the unbounded lognormal distribution. Using a bounded frequency distribution would solve this issue either, because then a high enough drawing frequency would produce INPC values from the upper bound of the distribution (instead of indefinitely large values for the lognormal case). This would be equivalent to a Fletcher-like parameterization, but using this arbitrary new upper bound of the distribution instead of the median, with,
50  in my opinion, no added value from the stochastic approach, but an additional complexity, increased computational cost, and lower reproducibility.

Conversely, do not I think using a very low drawing frequency makes sense. As mentioned above, it would mean that less representative INP concentrations are required for the parameterization to work, which defeats its whole purpose of better representing the observed INP range. In addition, using a very low frequency means that in a given grid cell the
55  INP concentration is random (possibly very high or very low) but does not change at all over the drawing period, which

is also very unrealistic. This would mean that a single grid point in the model domain will for example always freeze all droplets (or freeze none) during the whole frequency time, which is much larger than the advection time scale, during which different air masses or clouds can pass through this permanently high-INP (or low-INP) grid cell.

Thank you for your comment and the elaborate explanation of your train of thoughts. The sensitivity to the standard deviation, drawing frequency, and model resolution are indeed limitations of how the parameterization can be implemented in a model, but we consider them no fatal flaws in the approach itself. We now highlight the limitations we discovered in our analysis more explicitly in the respective sections (Sec. 3.4.2, 3.4.4, and 3.4.5) as well as in the conclusions and we added some more information in the outlook on how to further develop the parameterization scheme to meet these limitations. We would like to emphasize here that this paper represents a first attempt to use a random approach including variability of field measurements as a freezing scheme and should be seen as a novel concept and its analysis rather than a finished parameterization scheme to be used in other models. To highlight that we adapted the title of the paper to "The Chance of Freezing – A conceptional study to parameterize temperature-dependent freezing by including randomness of INP concentrations" and clarified this in the conclusions.

The sensitivities as presented in the paper (Sec. 3.4) illustrate fundamental issues one encounters when creating and implementing data-based parameterizations, like the bridging of spatial and temporal scales between observational and model resolution, the need to time-discretize deterministic (time-independent) schemes, etc. We do spell out some of these issues more explicitly in the revised paper.

To address some of the specific aspects as mentioned by the reviewer:

- It is not possible to increase the drawing frequency to higher values than the model time step, so we cannot test the model's sensitivity to higher drawing frequencies. However, we agree that ideally, the values should converge for higher drawing frequencies. Nevertheless, we do not expect the drawn INPC in all cloud grid cells to reach an INPC > Nc within relevant atmospheric time scales. The random drawing does indirectly introduce a time dependence of the freezing scheme (the time it takes until a high value is drawn). This is now further explained in Sec. 3.4.4.

- Keep in mind that we test the parameterization directly in an atmospheric model, where also other processes like advection, sedimentation, and deposition/sublimation are taken into account.

- We have added a simulation with the Fletcher parameterization to the manuscript (Sec. 3.3), which shows that almost no ice would be produced in the modeled cloud. Also, F23 does not require additional computational cost compared to a Fletcher implementation (see also our response to "Other comment 12") and we judge that there is an added value to the stochastical approach (in case the frequency distribution would, e.g., be limited by an upper boundary), since it does represent INPC observations more correctly (reflecting the variability of measurements).

- Using an adapted Fletcher parameterization (an upper bound parameterization as mentioned by the reviewer) would lead to a lot more ice in the simulated case as all grid points would instantaneously use the upper-bound INPC. This would not describe the freezing realistically (the time and spatial component from the random drawing gets

lost) and would again mean that the whole dataset is reduced to one parameterization line without accounting for the variability (which is another feature of the described parameterization scheme).

– We agree that very low drawing frequencies are not reasonable. The test of drawing once per hour was added to the manuscript to illustrate the convergence for low drawing frequencies.

– We emphasize (and now state more clearly in the manuscript, e.g., beginning of Sec. 2.1) that the random drawing of new INPCs does not necessarily mean that we assume that the aerosol population has changed, it could also represent a change in the present INP population due to activation of more INPs with time.

– Additionally, we think that the dependency on drawing frequency is what can be seen in observations: the higher the temporal resolution of time-averaged INPC measurements, the higher the chance to observe larger INPC values from timestep to timestep (see also Fig. 2 and its discussion).

2) The parameterization represents the global and annual INP variability for a given T, modeled as a random process, and reproduces this variability at all times and locations. However, I think that this observed variability arises largely from non-random local (temporal and spatial) factors. For example, INP concentrations are almost always much lower in the coastal Arctic (Wex et al., 2019, cited in the manuscript) than near the equator (for example the Cabo Verde data used in this study, Welti et al., 2018, also cited already). In the coastal Arctic, INP concentrations are also much lower in winter than in summer, and are probably even lower in the middle of the ice pack where no identified INP sources exist. Even in Cabo Verde, where the data used in this study originates, Welti et al., 2018 mention that the high time variability of INPs is partly explained by emission sources and meteorological contexts. The issue here is that the approach presented in the present study can produce very high INP concentrations where and when they should be very low, for example during the Arctic winter. It is true that the Fletcher parameterization which also be biased in conditions that deviate significantly from median conditions. However, in the parameterization of Fletcher, the bias on INP concentrations will be limited by the fact that the median INP was used to build the parameterization. In the present study, extremely high INPs can be predicted at sites where they should be extremely low, something that does not occur when using relationships like Fletcher. In low INP cases, this is made worse by the issue discussed above in Major comment 1).

We used the data sets from Welti et al. (2018) and Welti et al. (2020) because they are extensive marine data sets (thus representing a region with rather low INPCs), to extract the general features of the INPC RFD. As mentioned in the manuscript, the derived INPC RFD that we test in the parameterization is only loosely based on the dataset. The main goal of our study was to test the concept of random drawing from an INPC distribution instead of using fixed values only depending on, e.g., temperature, and one major result is that it is the rare, large INPCs that control the freezing in a cloud and not the median values. The specific definition of the RFD is not our main focus, but to investigate the effects of variables defining the RFD. We present sensitivity tests on how changes in the distribution affect the modeled cloud ice. These changed distributions reflect a wide spectrum of INPC RFDs. If the parameterization is to be used in a global

modeling context it would indeed be useful to derive different RFDs for different source regions of INP. That is stated as an outlook in the third paragraph of the conclusions.

We like to stress here again that using the Fletcher parameterization did not lead to any significant freezing in the simulated case (Sec. 3.3).

125 When it comes to the statement that the assumed INPC variability might be too high for the low-INP-environment case simulated here, Fig. 1 presents a comparison of the used distribution to observations in Ny-Ålesund (Wex et al., 2019). It shows that our distribution generally overlaps quite well with the data, especially in the temperature region (between

[Figure]

**Figure 1.** Relative frequency distribution spectra for INPC as a function of temperature. Black crosses show observations as reported in Wex et al. (2019) for Ny-Ålesund, vertical dashed lines indicate the temperature range of the simulated cloud in our study.

the gray dashed vertical lines) of the cloud that we simulate in this study.

Regarding the variability that we assume in our parameterization, we do conclude that this is plausible. Fig. 2(a) shows
130 exemplary observations at $\Delta t = 10$ s and Fig. 2(b) three corresponding drawing experiments from our RFD. The observations show a continuous variability between 0 and 300 $L^{-1}$ with one high peak at almost 1000 $L^{-1}$ independent of temporal or spatial factors of the measurement. The drawing experiments show similar behavior and mimic the observation well.

3) Unless I am mistaken, the parameterization of Fletcher predicts the total number of INPs, not just immersion mode
135 INPs (it is based on observations of primary cloud ice, when INPC=Ni the cloud ice crystal number). While the exact

(a) INPC observations done with SPIN and time resolution of 10 s. $T = 245$ K, $RH_w = 1.03$.

(b) Three experiments of drawing 100 times from our distribution at also $T = 245$ K: blue, green, and orange.

**Figure 2.** INPC observations with high time resolution in (a) and corresponding drawing experiments from the RFD in (b)

details of what is included in the STD control simulation are not clear, the present study seems to replace the Fletcher parameterization in the model, or a ice=f(T) relationship, by an immersion-mode-only parameterization. How are other INP processes (deposition, contact) parameterized in the model? This could also explain why the median INP=f(T) value on Figure 1. is much lower than in Fletcher, and, if other nucleation modes are completely ignored, why the new parameterization produces less ice than STD.

In Fletcher (1962), p. 239 it is stated: "The measurements were all made at approximately water saturation, in the presence of a fog of water droplets.", so the Fletcher-parameterization represents immersion and contact freezing only. Immersion freezing is the most relevant ice nucleation mode for the conditions in the modeled cloud and the case study presented. Contact freezing can be neglected since very little interstitial (not activated) aerosol was present in the case and given that these aerosols would still need to collide with the supercooled cloud droplets in a very stable and not very turbulent cloud the contribution expected is close to 0. Deposition freezing has the same limitation when it comes to interstitial aerosol and can additionally be neglected because of the temperature range of the case that is not favorable for deposition freezing. We have added this information in Sec. 2.2.

F23 only represents immersion freezing, no parameterization of other ice nucleation modes is used in the current model setup, and we do not use prognostic aerosol which makes aerosol-based parametrization schemes impossible.

In the STD case no interactive/prognostic freezing parameterization is used, but instead, Ni is kept on the constant value of 200/m3 where T < 0C and there are cloud droplets, which can be seen as a representation of processes that potentially create ice (immersion and contact freezing, SIP, but not e.g. deposition nucleation since cloud droplets are a condition). We have adapted the manuscript to emphasize these aspects more, see Sec. 2.2.

155  4) The abstract mentions that "reasonable ice masses" are produced by the model, but no comparison with observations of the new model is provided in the paper. Section 2.2 suggests that the original model underestimated cloud ice unless unrealistic INP concentrations were used, but the updated parameterization (F23) produces even less ice, so is it not degrading the model performance?

We added the observations to Fig. 3 in the manuscript and the new Sec. 3.2. Since we do not represent all ice hydrome-
160  teors or secondary ice processes, it is however difficult to compare the results with observations (in addition to the quite large uncertainties of the observations).

**1.2  Other comments**

1) The parameterization is said to be produced from observation data at Cabo Verde and on ships, but the comparison between observations and the parameterization is never shown. Can you add observed INP data on Figure 1 or add it as
165  a supplementary figure?

The distribution used in the parameterization is only loosely based on those observations. We used the data to estimate the width and approximate shape of our conceptual distribution since it is an extensive dataset, but it is possible to use other datasets and distributions in the parameterization as well. The focus of this conceptual study is not the exact distribution, but rather to test the approach of random drawing from an INPC distribution. We have emphasized this
170  aspect at the beginning of Sec. 2.1.

2) L.49-50: I think this is slightly misleading because some of the observed variability can actually be explained by these meteorological and aerosol parameters. Complex models do not reproduce the full observed variability of course, but that is true of any model of any quantity. In addition, as mentioned in the introduction, the models of Marcolli et al., 2007 and Wang et al., 2014 do parameterize some of this enhanced variability.

175  The drawbacks are of general nature and not specific to the mentioned models. To make this clearer we added a linebreak and rephrased the statement. We wrote that parameterizations **can** have these drawbacks and changed to "Most of them fail...".
Additionally, our parameterization is not limited to a specific type of aerosols, while Marcolli et al. (2007) represent only ATD and Wang et al. (2014) natural dust and black carbon.

180  3) L. 54: There is also an issue here, because the distribution presented in the paper does sample from the global variability of INPs, but is probably worse at representing the local (spatial and temporal) variability of INPs than aerosol-aware models, which for example will predict lower INPC over very remote regions where INPs are rare, and higher INPC over deserts where dust sources are plentiful.

185 For the focus of this study is not essential that the INPC RFD isn't based on local observations in the domain of the modeled case. The aim was to find a broadly applicable formulation that can easily be refined. It is a conceptual study of using log-normally distributed INPCs instead of fixed values. In the future, different INPC RFDs for different source regions could be used to account for this aspect (see third paragraph in conclusions).

4) L.73-74: Isn't this in contradiction with the discussion above of how variable INPs are in time and location? These observations used to build the parameterization should also be shown alongside the parameterization or in a supplementary
190 figure.

We don't see a contradiction here - we refer to there being similar distributions and that the variability is consistent. Additionally, we use that data only to find our conceptual INPC distribution.

5) Equation(1) – Why does the formula for $\mu$ gives negative values for $T > -10C$? Is there a typo here? I also think all the terms should be defined with units explained, is INPC supposed to be in #/m3?

195 There is no typo and we think it should be fine that $\mu < 0$ for $T > -10C$. Since it is the arithmetic mean of the **log**normal distribution, the log of the median of the INPCs (which can become negative for INPC < 1).
We extended the explanation in Sec. 2.1 and now include the units of variables in the equations. Thank you for spotting this.

6) L. 80 What do you mean by "normalizing the distribution"? What quantity is used to normalize it and why? A lognormal
200 distribution should also in theory already be normalized. I suppose the issue could be that equation 1 is missing a 1/INPC factor in order to be truly lognormal, unless the distribution was meant to be defined in units of dD/dlog(INPC) (since dlog(INPC)/dINPC = 1/INPC).

We normalize by the integral of values for all INPC at a specific T in order to obtain relative frequencies between 0 and 1. This is a constant factor depending on the definition of the INPC-field that the distribution is defined on.

205 7) L. 82 si>0 is only explicitly required for deposition freezing. For immersion freezing, there needs to be supercooled liquid droplets present (T<0., sw>0. and liquid droplets present). In practice, when the system is in equilibrium, si will always be positive where supercooled liquid exists, but this is not directly required for immersion freezing to occur.

We agree and removed the mentioning of this criterion in 2.1, 2.2, and 3.

8) L. 87-98. How are other nucleation modes treated? Even for immersion freezing only, INPs can also cause freezing of rain so why not also take into account the concentrations snow and other frozen precipitation when subtracting from INPC?

This section is exclusively about the new immersion freezing scheme, not the setup of the atmospheric model. In Sec. 2.2 we added information on why we only represent immersion freezing - it is the dominant mode for Arctic stratocumulus clouds relevant to the simulated case.
By considering only the freezing of cloud droplets (and not raindrops), we simplified the implementation and analysis. Fig. C1 shows that this is reasonable as Nc is much larger than Nr.
Since our simulations include only ice as frozen hydrometeors, it is only Ni we need to subtract. However, we now mention in the text above Eq.s 2 the necessity to include other frozen hydrometeors in the subtraction (Sec. 2.1).

9) L. 96-98: A new drawing from the distribution will regenerate INPs, so I think that if the drawing frequency is smaller or equal to the time step this cycling does not occur. In addition, does this mean that the drawing frequency, when larger than the time step, is also assumed to be the INP replenishment time in the cell by e.g. advection and resuspension?

Every new drawing from the INPC RFD tells us the INP concentration we assume for the current time step, it does not regenerate INPs because we subtract the amount of ice present in order to calculate the newly formed $\Delta$ Ni. This is to make sure not to regenerate INPs (and to get an infinite/additive amount of INPs) and is a compensation for not having prognostic INPs. We assume either replenishment of INPs in every timestep or time-dependent activation of INPs. The new drawing does not (necessarily) mean that the aerosol population has changed by, e.g., advection, but rather that a new representation of the INPs present is chosen. As mentioned before, it is not possible to have a drawing frequency smaller than the model time step.
We now clarify this in the beginning and end of Sec. 2.1.

10) L. 103-111 – Instead of discussing representativity with numbers of grid points for a given hypothetical cloud, I think it would be more clear to simply present this discussion as the representativity sample size of 1000 points.

We wanted to directly apply the representativity on the cloud to also stress the point of how the amount of ice formed is calculated from the drawing from the INPC distribution.

11) 2.1.1 and 2.1.2 Since the sampling of the distribution is done independently in each grid point, INP concentrations can be extremely different in a grid point and the ones directly next to it, but it is probably not the case in reality, where the temporal scale of an aerosol plume can be several hours or even days, and the spatial scale several 10s of kms or even 100s of kms (for example Saharan dust events).

We discuss the possibility to add autocorrelation in the third paragraph of the outlook, which would mimic such behavior.

12) L.128 – Is the doubling of computing time 1) for just F23 compared to Fletcher, 2) for the microphysical calculation only or 3) for the total simulation time? If 3), how can you explain such a large cost from only adding a random sampling calculation to the model?

It was for F23 versus STD (not Fletcher, but constant Ni, see more detailed description in the revised manuscript) and the total simulation time. The interactive ice nucleation schemes in MIMICA (everything apart from STD) include a more elaborate implementation of e.g. INP tracers (not used for F23), which leads to the large increase in total runtime. We agree that it is more informative to compare the runtimes of F23 to the Fletcher 62 parametrization and adapted the manuscript accordingly (now at the end of Sec. 2.1).

13) L. 130 Does the standard MIMICA (STD simulation) include Fletcher nucleation or is it with constant Ni, or something else? Description of the model in 2.2 indicates that Ni can vary since it is corrected if falling above 200/m3.

STD keeps a constant Ni of 200/m3. To clarify we have extended the description of this simulation in 2.2. Ni in general can change in the model (and thus fall below or above 200/m3) due to, e.g., turbulent transport or precipitation of ice crystals (we added this information at the end of 2.2).

14) L. 144 – Can secondary ice production in the cloud explain these ice biases, or has this been ruled out?

Secondary ice processes would not be enough to increase the IWP to the observed values (similar to, e.g., the Fletcher simulation now presented in 3.3). In the new section 3.2, we now mention how secondary ice processes modify the modeled IWP, with respect to F23 (factor 2 to 3) following Sotiropoulou et al. (2021). However, this would not be enough to increase the IWP using Fletcher (1962) close to the observations (adapted Fig. 3 in manuscript).

15) L. 162 – How was graupel and snow removed and can clouds precipitate or be removed by another process? Can this removal cause issues, for example, if there is no precipitation in the model have you checked that the cloud droplets and ice crystals do not grow to unrealistic sizes?

The model does not include the formation of snow or graupel in our simulation setup (no conversion processes to these hydrometeor classes are represented). We now specify at the end of 2.2 that ice crystals can continue to grow by deposition and precipitate. Also, raindrops exist in the simulations and can precipitate.

See Fig. 3 for the average mass of simulated ice crystals. It is approximately $3.6 \cdot 10^{-8}$ kg = 36 $\mu$g. This is a realistic
mass (see, e.g., Vázquez-Martín et al., 2021).

16) 2.2 and l. 171-172 – Can you explain more explicitly what are the differences between STD and F23, is this just the ice
nucleation or are there other changes to how cloud ice is handled? Is graupel and snow also removed from STD?

We now clarify at the beginning of 3.1 that the only difference between F23 and STD is the parameterization of ice
formation and clarify at the end of 2.2 that graupel and snow are excluded from all simulations presented in the paper
(including STD).

17) 2.2 and Figure 3. - You mention that the issue in STD is the difficulty in reproducing observed cloud ice, but the IWP is
even lower in F23, is this not degrading model results?

Using the observation-based INPC distribution does increase the discrepancy between observed and simulated IWP
compared to STD. The sensitivity test on the RFD standard deviation (Fig. 7 in the revised manuscript) demonstrates
that the parameterization is flexible and could be forced to match the observed IWP by, e.g., increasing the width of the
INPC RFD.
In Sec. 2.2 we mention that the use of a different interactive ice nucleation scheme, e.g. an active site scheme following
Ickes et al. (2017), requires unrealistically ice-active INPs, this does not refer to STD. We adapted the text there and
hope it is now clearer. Additionally, we have added explicit comparisons of F23 and STD to observations (Sec. 3.2) and
explain the respective discrepancies to the observations there.

18) L. 181 - Can you provide a number estimate or figure showing this latent heat or temperature increase in the model?

To address this point, we added plots of the mixing ratio tendencies due to condensation/evaporation and sublima-
tion/deposition to appendix B.

19) L. 183- Can you also plot the water vapor column to show this gas-phase increase? The water budget should be balanced
and it should be possible to show this shifting of water around phases directly.

See response to "Other comment 18" above.

20) L. 185 – Unfortunately, this increase with time is probably at least partly caused by the issues with the parameterization
discussed above, because with enough time, INPC values large enough to freeze any supercooled droplet concentration

will be sampled from the distribution. It seems that when discussing the sensitivity to drawing time the slope of the

290 increase is indeed larger for smaller drawing times and the gap between experiments grows during the simulation.

We have added these comments to the manuscript in 3.1 and 3.4.4.

21) L. 186 Can you also calculate here directly this effective cloud ice crystal mass from model results of ice crystal numbers and ice mass concentrations?

After closer inspection, it turned out (from calculating IWP/ICNB for the respective comparison runs of STD and F23)

295 that the average ice mass of one crystal follows a similar curve for both STD and F23 (see Fig. 3). We removed this from the manuscript.

[Figure]

**Figure 3.** Average mass of an ice crystal in STD and F23 comparison run. From IWP/ICNB.

22) L. 231- 232 - This is very interesting and probably true for other parameterizations. For this reason, a parameterization only based on a median of INP observations will probably underestimate primary ice formation and some sort of correc­tion might be necessary to consider short term variability and the likelihood of efficient INPs deviating from the median

300 values. However, I am not sure of how applicable to other approaches this is. For example, earlier simple parameteri­zations like the one from Fletcher were I think directly based on observed ice crystal numbers, and did not suffer from this issue directly. And I think that some of the more complex aerosol-aware parameterizations will assume that these efficient INPs are quickly removed from the atmosphere and will not continue to nucleate ice later.

This is a valid point. The first part clearly reflects what we wanted to accomplish with this study: that it is not enough to

305     represent INPs by the median of observations. We now state this more clearly in the revised outlook (second paragraph). See also the newly added Sec. 3.3 that shows that much less ice is produced when using the Fletcher parameterization in the modeled cloud.

Note that more complex aerosol-aware parameterizations will not assume that efficient INPs are quickly removed from the atmosphere since most models do not treat INPs prognostically, which basically means that the efficient INPs can
310     appear again in every time step.

23) L. 260-261 – Is there a quantitative estimate of INPC variability at very short time scales (a few minutes and meters) in these references? INP are often observed at very low resolutions (very few observation sites, daily resolution or lower) so I am not sure how large this small-scale variability is, compared to the large-scale variability used to build the parameterization.

315 We double-checked the citation and it says in Bigg (1961): "... in which the concentration of nuclei sometimes increased by as much as a factor of 100 in less than an hour. " (p. 9).

As to the other aspects, see our response to "Major comment 2", including Fig. 2. To avoid confusion, we removed this sentence.

**2    Response to review 2**

320 **2.1    Major comments**

1) My most major concern about the dataset used is that it only uses surface-based INP concentrations which may not be representative of the cloud-layer (see e.g. Creamean et al. 2018, Griesche et al. 2020). For a decoupled cloud system such as the one that is actually the focus of this study, the surface-level INPs may not impact the Arctic low-level clouds of interest. Furthermore, these observations were made over Cape Verde, while the parameterization is being tested in
325 the Arctic, which has very different INP abundance and composition. Assuming this parameterization might also be applicable to large-scale climate models, it has been shown that the vertical structure of INPC can play an important role in simulating the Arctic and how it changes in the future is especially sensitive to the vertical structure of INPC in a large-scale model (Tan et al. 2022). Please discuss the limitations of the assumptions of the framework in the context of the surface observations that were used and how they may influence the simulation of cloud properties and how they
330 may also potentially influence future climate projections if the scheme can be applied to climate models.

We thank reviewer 2 for this valuable comment. We agree that surface-based measurements of INPCs do not necessarily represent INPC at the cloud level. We assume here that the boundary layer is well-mixed and that our conceptual distribution is valid also at the cloud level. We would like to highlight that this is an outstanding question in the field and true closure is still missing.

335 However, our study does not aim at offering the correct INPC distribution, but to test the general idea of representing INPs through a distribution rather than fixed values (adapted at beginning of 2.1 in the manuscript). That is why we consider the mentioned limitations to be only of secondary concern.

Regarding the difference between INPCs in Cabo Verde and the Arctic, see the second to last paragraph of our response to reviewer 1's "Major comment 2" and Fig. 1, where we show that INPCs are comparable.

340 We mention the problem of changing INPC distributions in future climate in the conclusions (in the third paragraph). Regarding the influence of different INPC distributions on cloud properties, see the sections on the sensitivity tests of the median and standard deviation of the RFD (Sec. 3.4.1 and 3.4.2).

2) It is curious that the ASCOS observations are not directly plotted in the figures. Could the authors add the observations for a more direct quantitative comparison with their results instead of qualitatively referring to previous papers?

345 This is now done. See the adapted Fig. 3 in the manuscript and the new Sec. 3.2, as well as our response to reviewer 1's "Major comment 4".

3) I'm unclear about why the sensitivity test for the sampling frequency is designed the way it is. In particular, starting on lines 264-267: "Freezing events still occur at every time step, but within the respective time period (e. g. five seconds or 60 minutes), the INPC is constant at one grid point for the time period. If the temperature at the grid point changes

350 before the completion of the time period, a new INPC is drawn earlier." Because a new INPC will be drawn earlier if the temperature of the grid point changes before the completion of the time period, then that should mean that each of the sensitivity tests will draw a new INPC before the specified frequency. It therefore seems to me that specifying the frequency in the sensitivity test is not useful since a new INPC can be drawn before the specified frequency. So in the end, are the simulations really comparing the effect of sampling frequency? Could the authors clarify why the sensitivity

355 test was designed this way?

We have changed this sensitivity test and now omit the drawing of new INPC in case the temperature changes before the completion of the time period (Sec. 3.4.4).

4) If the various sensitivity tests are not identically initialized, then if an INPC is drawn before the time period specified for the sensitivity test ends, then the comparison of the different sampling frequencies are not directly comparable. However,

360 if all simulations are initialized

See reply to the previous comment. We believe this comment was an artifact?

5) It's not clear to me why the authors deem 300 random draws to be the threshold for a reproducible prediction of the RFD. From Fig. 2, it appears that the mean root-mean square error (RMSE) seems to plateau closer to 1000 draws, when the mean RMSE is < 5. This is substantially larger than 300 draws and more than the 9216 grid points they quoted with the same temperature. Could the authors explain how they arrived at this conclusion in Section 2.1.2?

We tried to determine the threshold from statistical tests, but they did not give conclusive results. Our reasoning is that at n=300, the slope of the mean RMSE becomes approximately linear and added "(linear slope)" in 2.1.2. However, we agree that this value is somehow arbitrary.

**2.2 Minor comments**

1) The acronym ASCOS should be spelled out when it first appears on line 66.

Thank you. We have added it.

2) Line 97: With regard to Solomon et al. 2015, I think you mean INP "recycling" (and not "cycling").

Thank you. We have corrected it.

3) How exactly are the INPC drawn from the distribution? Does the subroutine use a random number generator?

Yes, we use the Fortran routine random_number and random_seed it with a number that depends on the MPI-process ID and the current time. Random_number returns a pseudo-random number from the uniform distribution between 0 and 1, which we use together with the cumulative frequency distribution of the INPCs to find the matching INPC.

4) Line 233: "on" should be replaced with "at".

Corrected, thank you.

5) In Fig. 3, I can't find anywhere in the manuscript that explains how the different simulations in the ensemble members differ. Please explain.

We explained that in the appended description of the model, but now moved it into Sec. 2.2. To clarify, we also added it at the end of Sec. 3.

6) Lines 233-236: This discussion about secondary ice production (SIP) is however not relevant to the discussion of Fig. 8 since the appendix states that MIMICA does not represent SIP (line 479).

The discussion here is to emphasize which implications a high Ni (due to the adapted INPC RFD) could have in other simulations. We have adapted the manuscript to make this clearer.

7) Some of the explanations are speculative although plausible (e.g. pertaining to discussion starting on line 177). Can the tendencies of the associated processes be displayed somewhere to substantiate this discussion?

See our response to reviewer 1's "Other comment 18": We have now added plots of the mixing ratio tendencies due to condensation/evaporation and sublimation/deposition to appendix B.

8) How are other modes of ice nucleation represented (if at all)?

Only immersion freezing is parameterized as it is assumed to be the most important ice nucleation mechanism for stratiform Arctic mixed-phase clouds. See our response to reviewer 1's "Other comment 8" and the adapted Sec. 2.2.

**References**

Bigg, E. K.: Natural Atmospheric Ice Nuclei, Sci. Prog., 49, 458–475, http://www.jstor.org/stable/43425202, 1961.

Fletcher, N.: The physics of rainclouds, Cambridge University Press, 1962.

Ickes, L., Welti, A., and Lohmann, U.: Classical Nucleation Theory of Immersion Freezing: Sensitivity of Contact Angle Schemes to Thermodynamic and Kinetic Parameters, Atmospheric Chem. Phys., 17, 1713–1739, https://doi.org/10.5194/acp-17-1713-2017, 2017.

Marcolli, C., Gedamke, S., Peter, T., and Zobrist, B.: Efficiency of Immersion Mode Ice Nucleation on Surrogates of Mineral Dust, Atmos. Chem. Phys., 7, 5081–5091, https://doi.org/10.5194/acp-7-5081-2007, 2007.

Sotiropoulou, G., Ickes, L., Nenes, A., and Ekman, A. M. L.: Ice Multiplication from Ice–Ice Collisions in the High Arctic: Sensitivity to Ice Habit, Rimed Fraction, Ice Type and Uncertainties in the Numerical Description of the Process, Atmospheric Chem. Phys., 21, 9741–9760, https://doi.org/10.5194/acp-21-9741-2021, 2021.

Vázquez-Martín, S., Kuhn, T., and Eliasson, S.: Mass of Different Snow Crystal Shapes Derived from Fall Speed Measurements, Atmospheric Chem. Phys., 21, 18 669–18 688, https://doi.org/10.5194/acp-21-18669-2021, 2021.

Wang, Y., Liu, X., Hoose, C., and Wang, B.: Different Contact Angle Distributions for Heterogeneous Ice Nucleation in the Community Atmospheric Model Version 5, Atmospheric Chem. Phys., 14, 10 411–10 430, https://doi.org/10.5194/acp-14-10411-2014, 2014.

Welti, A., Müller, K., Fleming, Z. L., and Stratmann, F.: Concentration and Variability of Ice Nuclei in the Subtropical Maritime Boundary Layer, Atmospheric Chem. Phys., 18, https://doi.org/10.5194/acp-18-5307-2018, 2018.

Welti, A., Bigg, E. K., DeMott, P., Gong, X., Hartmann, M., Harvey, M., Henning, S., Herenz, P., Hill, T., Hornblow, B., Leck, C., Löffler, M., McCluskey, C., Rauker, A. M., Schmale, J., Tatzelt, C., van Pinxteren, M., and Stratmann, F.: Ship-Based Measurements of Ice Nuclei Concentrations over the Arctic, Atlantic, Pacific and Southern Ocean, Atmospheric Chem. Phys., pp. 1–22, https://doi.org/10.5194/acp-2020-466, 2020.

Wex, H., Huang, L., Sheesley, R., Bossi, R., and Traversi, R.: Annual Concentrations of Ice Nucleating Particles at Arctic Station Ny-Ålesund, PANGAEA, https://doi.org/10.1594/PANGAEA.899696, 2019.

---

## Author Response (AR2)

**The Chance of Freezing - A conceptional study to parameterize temperature-dependent freezing by including randomness of INP concentrations**

Hannah C. Frostenberg[1], André Welti[2], Mikael Luhr[3], Julien Savre[4], Erik S. Thomson[5], and Luisa Ickes[1]

[1]Department of Space, Earth and Environment, Chalmers University, Gothenburg 41296, Sweden
[2]Finnish Meteorological Institute, Helsinki 00101, Finland
[3]former Department of Meteorology, Stockholm University, Stockholm 10691, Sweden
[4]Meteorological Institute, Faculty of Physics, Ludwig-Maximilians-Universität, Munich 80333, Germany
[5]Department of Chemistry and Molecular Biology, University of Gothenburg, Gothenburg 41296, Sweden

We thank both reviewers for the constructive feedback in this second round of reviews. Below we copied the reviewers' comments in black and added our responses in blue.

**1 Response to referee 1 (report #2)**

**General comment**

I thank the authors for the care they took in answering my earlier comments as R1 in the previous review. Requests for clarification and technical comments in particular were very well adressed.

I think the manuscript is now stronger and more clearly presented as a first exploration of a novel concept in parameterizing INPs in models. However I think this last point could still be improved: the authors stress in their answer that "this paper represents a first attempt to use a random approach [...] and should be seen as a novel concept and its analysis rather than a finished parameterization scheme to be used in other models". I think such an exploratory study is very relevant for ACP, but that this point still needs to be stressed as explicitly as in this citation in the abstract. Also in the response: "The main goal of our study was to test the concept of random drawing from an INPC distribution instead of using fixed values only depending on, e.g., temperature, and one major result is that it is the rare, large INPCs that control the freezing in a cloud and not the median values". This could be in my opinion also present as-is at the end of the introduction when presenting the objectives of the study. It would be better in order to indicate more clearly the limitations of this approach and how this work should be extended in future research.

Thank you for this feedback. To state limitations and exploratory character of the study more clearly, we added similar statements to the abstract and at the end of the introduction as suggested by the reviewer (see revised manuscript).

**1.1 Detailed comments**

20    1) "It is not possible to increase the drawing frequency to higher values than the model time step". It would be possible to decrease the model time step and use this as a new baseline, but I agree that the additional computing cost makes this unreasonable to ask for here. Maybe you should note here that for this reason, changing the model time step could have an effect on the parameterization.

We added this limitation and implication of the time step to the manuscript in section 3.4.4 (see revised manuscript).

25    2) "However, we agree that ideally, the values should converge for higher drawing frequencies". I also think this should be said explicitly in the text, and that this is something that future work should investigate.

We added this now in section 3.4.4 and in the conclusions (see revised manuscript).

3) Using the Fletcher parameterization leads to even lower ice formation than the new parameterization. But is this the only difference between the schemes? What would be the difference between this scheme and a Fletcher parameterization

30    tuned up to produce more ice? The discussion could mention that future work could investigate how the parameterization's behaviour differs from a time-independent one, other than just the amount of ice which could always be adjusted up or down in models. The added value of the new scheme could be more clearly revealed from a detailed comparison with a simpler adjusted scheme.

Unfortunately, we did not investigate the difference between this scheme and a tuned Fletcher parameterization (we

35    agree that this would be an interesting test for further studies). One question here is how much the Fletcher scheme would need to be tuned and if that is still a realistic representation. We can try to make some assumptions based on some of the simulations that we did. In Fig. 5 one can see the difference in the new scheme due to a shift in the median of the RFD. To achieve a more realistic IWP one would need to shift the median by at least a factor of three. Since the Fletcher scheme is a lot lower compared to the standard of the new parameterization (F23), it would need to be tuned a lot more

40    than that, probably at least by one order of magnitude.
Following the reviewers suggestion we added this point as outlook on further research at the end of Section 3.3 (see revised manuscript).

4) It's still not clear why a lognormal distribution of INPC would require additional normalization for sampling. Drawing from a lognormal distribution, while not completely straightforward in FORTRAN, does not require some special

45    renormalization (see for example https://numpy.org/doc/stable/reference/random/generated/numpy.random.lognormal. html for an implementation of a lognormal draw in python, and for a normal distribution in FORTRAN an implementation of the Box-Muller algorithm here https://masuday.github.io/fortran_tutorial/random.html). You should consider attaching the implementation and especially the sampling code to the article to make this work more easily reproducible.

50    The distribution is normalized to give a cumulative probability of 1 at each temperature, independent of the discretization of the INPC field (size of the bins). The normalization factor is a constant, depending on the definition of the INPC field. We added a clarification on this in Section 2.1 (see revised manuscript).

5) "Note that since large INPC and the chance to draw this large INPC determine the ice in the cloud, this introduces an indirect time-dependency of the scheme. The resulting ice in the cloud depends indirectly on the time until a large INPC
55    value is drawn at a specific grid point. Increasing the drawing frequency leads to increases in the ice variables that do not appear to converge" You could note here that it might be possible to correct for this time scale dependence by introducing an additional correction parameter based on the drawing frequency in future implementations.

Good point. We added this in Section 3.3.4 (see revised manuscript).

**2    Response to referee 2 (report #1)**

60    I thank the reviewers for their item-by-item responses. Regarding the first and most major comments, although they have been answered, I do not see where they were addressed in the manuscript itself along with relevant references.

For clarity we restate the first and most major comment to which Referee 2 refers:

*My most major concern about the dataset used is that it only uses surface-based INP concentrations which may not be*
65    *representative of the cloud-layer (see e.g. Creamean et al. 2018, Griesche et al. 2020). For a decoupled cloud system such as the one that is actually the focus of this study, the surface-level INPs may not impact the Arctic low-level clouds of interest. Furthermore, these observations were made over Cape Verde, while the parameterization is being tested in the Arctic, which has very different INP abundance and composition. Assuming this parameterization might also be applicable to large-scale climate models, it has been shown that the vertical structure of INPC can play an important*
70    *role in simulating the Arctic and how it changes in the future is especially sensitive to the vertical structure of INPC in a large-scale model (Tan et al. 2022). Please discuss the limitations of the assumptions of the framework in the context of the surface observations that were used and how they may influence the simulation of cloud properties and how they may also potentially influence future climate projections if the scheme can be applied to climate models.*

What may not have been clear in our previous response is that the discussion which is initiated by the referee's comment,
75    namely the connection between ground-level measurements and cloud-level INP etc., introduces significant open questions

in the field. However, they are largely beyond the scope of this study, which was to investigate how we might incorporate variability and a distribution of INP occurrence into a cloud-resolving model. For this exercise, the width of the distribution and the temperature-dependent change in INPC are much more relevant than the absolute INPC that the reviewer is concerned about. We simply chose a representative INP distribution, and also showed in the previous response that it is robust for both Arctic measurements and the equatorial measurements from which it was extracted. It is for these reasons the referee does not find much altered text directly addressing the comment. We hope our re-emphasis of the "conceptual character" of our study serves to help clarify this point.

---

## Author Response (AR3)

**The Chance of Freezing - A conceptional study to parameterize temperature-dependent freezing by including randomness of INP concentrations**

Hannah C. Frostenberg[1], André Welti[2], Mikael Luhr[3], Julien Savre[4], Erik S. Thomson[5], and Luisa Ickes[1]

[1]Department of Space, Earth and Environment, Chalmers University, Gothenburg 41296, Sweden
[2]Finnish Meteorological Institute, Helsinki 00101, Finland
[3]former Department of Meteorology, Stockholm University, Stockholm 10691, Sweden
[4]Meteorological Institute, Faculty of Physics, Ludwig-Maximilians-Universität, Munich 80333, Germany
[5]Department of Chemistry and Molecular Biology, University of Gothenburg, Gothenburg 41296, Sweden

We thank the reviewer for the constructive feedback in this third round of reviews. Below we copied the reviewer's comment in black and added our response in blue.

**1    Response to referee 2 (report #1)**

The reviewer thanks the authors for clarifying the scope of their work. The points that the reviewer mentions relating to the limitations of the study should be included in the manuscript.

We have now added section 3.5, covering limitations and challenges of our study, to the manuscript.